# Dynamical phases of short-term memory mechanisms in RNNs

**Bariscan Kurtkaya** [* 1 2]  **Fatih Dinc** [* 2 3 4]  **Mert Yuksekgonul** [5]  **Marta Blanco-Pozo** [2 6]  **Ege Cirakman** [2]
**Mark Schnitzer** [2 6 7]  **Yucel Yemez** [1]  **Hidenori Tanaka** [† 8 9]  **Peng Yuan** [† 10]  **Nina Miolane** [† 3]

## Abstract

Short-term memory is essential for cognitive processing, yet our understanding of its neural mechanisms remains unclear. Neuroscience has long focused on how sequential activity patterns, where neurons fire one after another within large networks, can explain how information is maintained. While recurrent connections were shown to drive sequential dynamics, a mechanistic understanding of this process still remains unknown. In this work, we introduce two unique mechanisms that can support this form of short-term memory: *slow-point manifolds* generating direct sequences or *limit cycles* providing temporally localized approximations. Using analytical models, we identify fundamental properties that govern the selection of each mechanism. Precisely, on short-term memory tasks (delayed cue-discrimination tasks), we derive theoretical scaling laws for critical learning rates as a function of the delay period length, beyond which no learning is possible. We empirically verify these results by training and evaluating approximately 80,000 recurrent neural networks (RNNs), which are publicly available for further analysis[1]. Overall, our work provides new insights into short-term memory mechanisms and proposes experimentally testable predictions for systems neuroscience.

---

[*]Equal contribution † Supervision. [1]Koc University, Turkey [2]CNC Program, Stanford University, Stanford, USA [3]Geometric Intelligence Lab, UC Santa Barbara, Santa Barbara, USA [4]Kavli Institute for Theoretical Physics, UC Santa Barbara, Santa Barbara, USA [5]Computer Science, Stanford University, Stanford, USA [6]James H. Clark Center for Biomedical Engineering & Sciences, Stanford University, Stanford, USA [7]Howard Hughes Medical Institute, Stanford University, Stanford, USA [8]Phi Lab, NTT Research, Sunnyvale, USA [9]Center for Brain Science, Harvard University, Cambridge, USA [10]Institute for Translational Brain Research, Fudan University, Shanghai, China. Correspondence to: Fatih Dinc <fdinc@ucsb.edu>.

*Proceedings of the 42ⁿᵈ International Conference on Machine Learning*, Vancouver, Canada. PMLR 267, 2025. Copyright 2025 by the author(s).

[1]https://github.com/fatihdinc/dynamical-phases-stm

## 1. Introduction

Learning to respond based on delayed information is a fundamental challenge in both neuroscience and machine learning (Constantinidis & Klingberg, 2016; Bengio et al., 1994). In machine learning, early studies in the 1990s identified this difficulty as the problem of learning long-term dependencies, particularly in recurrent neural networks (RNNs) trained on tasks requiring information retention across time (Bengio et al., 1994). Despite significant progress in neural network architectures, this issue remains central to designing systems capable of effective temporal reasoning.

To address long-term dependencies, machine learning researchers developed specialized architectures such as Long Short-Term Memory (LSTM) networks, Gated Recurrent Units (GRUs), and, more recently, Transformers (Hochreiter, 1997; Chung et al., 2014; Vaswani, 2017). These models introduced mechanisms that enhance memory retention, *e.g.*, by learning neuronal timescales. In contrast, biological neurons communicate through action potentials and are governed by biophysical parameters, such as membrane time constants, that determine the decay of activity over time (Jonas et al., 1993; McCormick et al., 1985; Miles, 1990; Isokawa, 1997; Geiger et al., 1997). This fundamental difference raises a key question: how do biologically inspired neural systems learn to associate events separated in time?

This question has regularly been studied under a central topic in neuroscience: short-term memory, a fundamental cognitive function that enables organisms to temporarily store and manipulate information. This process is essential for adaptive behavior and is disrupted in several neurological and psychiatric conditions such as Alzheimer's disease (Bondi et al., 2017), schizophrenia (Lee & Park, 2005) and post-traumatic stress disorder (Scott et al., 2015) among others. Yet, while short-term memory is a central cognitive function, the mechanisms that support it remain poorly understood in systems neuroscience (Serences, 2016).

In the mammalian brain, short-term memory is supported by a dynamic network spanning cortical and subcortical regions such as the prefrontal, parietal, and sensory cortices, and the hippocampus (Todd & Marois, 2004; Curtis & D'Esposito, 2003; Harrison & Tong, 2009; Gnadt & Andersen, 1988; Baeg et al., 2003; Cavanagh et al., 2018; Meyers

et al., 2008). Theories explaining this core process can be broadly grouped into three levels. At the synaptic transmission level, memory may be stored in "activity-silent" states, maintained by transient synaptic changes induced by plasticity (Mongillo et al., 2008; Stokes, 2015; Kozachkov et al., 2022; Masse et al., 2019; Brennan & Proekt, 2023). At the neuronal activity level, persistent spiking has been linked to memory maintenance during delay periods (Gnadt & Andersen, 1988; Barash et al., 1991; Curtis & D'Esposito, 2003; Baddeley & Hitch, 1974) (Fig. 1A). Finally, at the population level, short-term memory has been associated with sequential patterns of neural activation (Baeg et al., 2003; Harvey et al., 2012; Rajan et al., 2016; Wang, 2021) (Fig. 1B).

In this work, we study mechanisms that can support memory maintenance at the final population level (Fig. 1B-C). Increasing evidence suggests that memory stability and robustness can emerge from a dynamic, distributed neural population activity and is governed by underlying attractor states (Cavanagh et al., 2018; Spaak et al., 2017; Meyers et al., 2008; Stroud et al., 2024; Brennan & Proekt, 2023). Yet several *theoretical*, and empirically relevant, questions remain unresolved: **Q1:** What mechanisms support memory maintenance through sequential neural activity? **Q2:** What determines which mechanism is favored under different memory tasks or training conditions? **Q3:** And how do these learned mechanisms vary with the delay duration, *i.e.*, the time during which the information must be held in memory?

We set out to answer these questions through a computational theory perspective, generating predictions for in-vivo experiments that we then test in-silico. To achieve this, we trained approximately 80,000 recurrent neural networks (RNNs) to perform classical system neuroscience experiments designed to assess short-term memory capabilities, while also studying simple toy models and low-rank RNN models as they performed a simpler short-term memory task (Fig. 1D-E). Our contributions can be summarized as follows: (i) Using interpretable dynamical system models stripped down to their most essential components for solving a delayed activation task (Fig. 1D-E), we illustrate how RNNs can develop distinct strategies during learning, depending on the task and optimization parameters (Figs. 2, 3, and S1). (ii) We show how changing one of the simplest tasks in a trivial manner, *e.g.*, by adding a post-reaction period, can qualitatively change the learned short-term memory mechanisms. This finding is particularly relevant as a cautionary tale for the systems neuroscience community modeling animal behaviors by drawing comparisons to RNNs (Mante et al., 2013) (Fig. 3). (iii) Utilizing

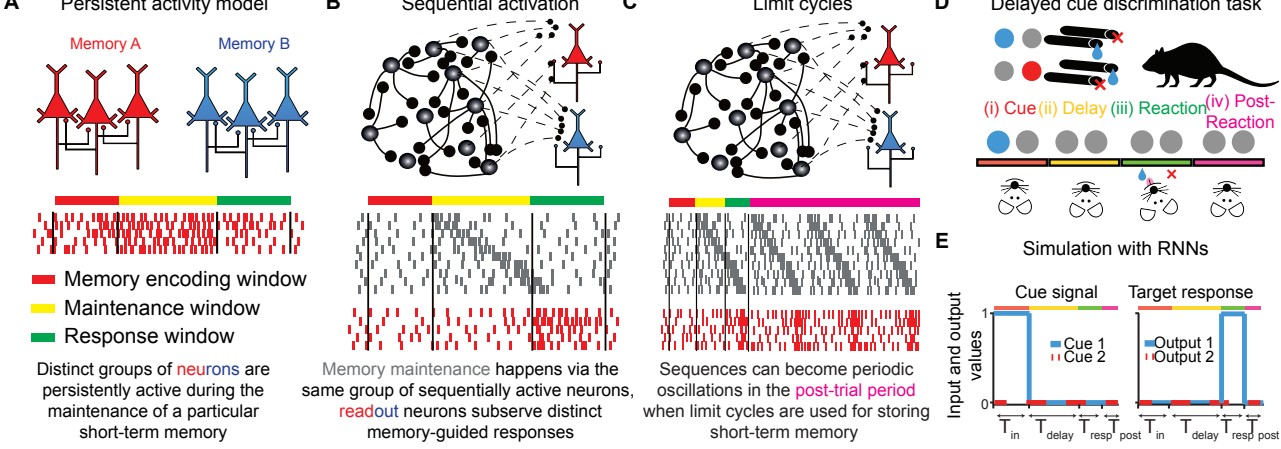

*Figure 1.* **Studying short-term memory mechanisms with biologically motivated neural network models. A-C** A visualization of short-term memory maintenance in neural activations. **A** Earlier models predict that fixed and persistent activities of a subset of neurons during the delay period could encode the content corresponding to distinct memories (red vs blue populations) (Fuster & Alexander, 1971; Atkinson & Shiffrin, 1968). **B** Memory contents can also be stored within a sequentially firing neuron population (Rajan et al., 2016), where distinct readout neurons decode distinct memories. **C** In this work, we show that limit cycles, *i.e.*, periodic orbits, can locally approximate the memory mechanism in panel (**B**). However, if left undisturbed, limit cycles may result in spurious periodic responses in the post-reaction window (magenta). **D-E** An illustration of the delayed cue-discrimination task and its simulation with RNNs. **D** The task includes two correct cue-reaction matches (blue-left, red-right) and consists of four periods: (i) Cue period, where one of the two cues is presented; (ii) Delay period, with no expected reaction; (iii) Reaction period, where the cue class needs to be outputted; and (iv) an optional post-reaction period at the end of the trial. **E** We designed an equivalent task for RNNs, in which two input channels are mapped to the output channels and for a given trial only a single input is provided in the form of a rectangular pulse of duration $T_{\text{in}}$. RNNs can be trained to follow the stages (i-iv) similar to the animals in their target responses. When the cue period is omitted, we refer to this task as delayed activation task.

analytical models, we demonstrate theoretical scaling laws and synthesize a phase diagram of mechanisms for solving the task, which is a function of the task properties and optimization parameters (Figs. S2 and 4**G**). (iv) Through a large-scale study of RNNs[2], we extract the same scaling laws in RNNs trained to perform delayed cue-discrimination tasks, which are quantitatively consistent with our theoretical explorations within error bars. We show that RNNs have little to no trouble learning a modified task with no delayed output, providing further evidence that training biologically inspired neural networks to delay their outputs is fundamentally difficult (Figs. 4, S3, S4, S5 and S6).

## 2. Background

Systems neuroscience often utilizes animal models, *e.g.*, mice, rats, birds, and flies, to study cognitive functions including short-term memory (Serences, 2016). Alternatively, it is often possible to design artificial models such as RNNs, which can be trained on observed neural activities or tasks that are traditionally performed by animals in laboratories (Mante et al., 2013; Sussillo & Barak, 2013; Yang et al., 2019; Masse et al., 2019; Dubreuil et al., 2022). Then, once RNNs are trained, their computation can be reverse-engineered to investigate the nature of the neural computation performed by biological networks, generating hypotheses that can later be tested with experimental procedures (Walker et al., 2019; Finkelstein et al., 2021). Consequently, RNNs have recently become a key component of computational neuroscience studies (Dubreuil et al., 2022; Valente et al., 2022; Finkelstein et al., 2021; Sussillo & Barak, 2013).

Here, we consider a family of RNNs that are biologically relevant and whose parameters can be interpreted as functional connections between neurons (Perich & Rajan, 2020; Perich et al., 2021). Specifically, we study RNNs defined by the following equation:

$$\tau \dot{r}(t) = -r(t) + \tanh(Wr(t) + W^{\text{in}}u(t) + b + \epsilon), \quad (1)$$

where $r(t) \in \mathbb{R}^N$ refers to the firing rates of $N$ neurons, $u(t) \in \mathbb{R}^{N_{\text{in}}}$ pre-defined $N_{\text{in}}$-dimensional inputs, $W \in \mathbb{R}^{N \times N}, W^{\text{in}} \in \mathbb{R}^{N \times N_{\text{in}}}$ weight parameters with $b \in \mathbb{R}^N$ corresponding bias terms, and $\epsilon$ some noise that is, in practice, sampled i.i.d. from a Gaussian distribution. The output is taken as a projection of the neural activities as $\hat{o}(t) = f(W^{\text{out}}r(t) + b_{\text{out}})$, where $f(\cdot)$ is either identity or sigmoid function in this work (see below)

---

[2]The dataset is publicly available through https://doi.org/10.5281/zenodo.15529757. To obtain this dataset, we used several computers with NVIDIA RTX 3090 GPUs or equivalent, which roughly amounts to 230 kg $CO_2$ emission, 930 km driven by an ICE car, 115 kg coal burned, 4 tree seedlings sequesting carbon for 10 year as computed via https://mlco2.github.io/impact/.

and $W^{\text{out}} \in \mathbb{R}^{N_{\text{out}} \times N_{\text{rec}}}$ and $b_{\text{out}} \in \mathbb{R}^{N_{\text{out}}}$ for an $N_{\text{out}}$-dimensional output. For notational simplicity, we assume $\epsilon = 0$ until our empirical discussions in Section 3.4.

*RNN models of sequential activity*–Recent work has identified the generation of neural sequences as a potential key component of short-term memory (Rajan et al., 2016). This *distributed* sequence can be used to store the short-term memory content and explicitly keep track of time, which contrasts with earlier theories of short-term memory maintenance relying on the persistent activities of some neurons (Fuster & Alexander, 1971; Atkinson & Shiffrin, 1968) (Fig.1**B**). Previous work has provided insights into how these sequences are generated by examining neuronal-level mechanisms (Rajan et al., 2016; Sommer & Wennekers, 2005; Laje & Buonomano, 2013), often focusing on the learned connectivity structures (Rajan et al., 2016; Laje & Buonomano, 2013). However, the *population-level* causal mechanisms underlying neural computation, which could be investigated through latent dynamics, remain largely unexplored.

To study the latent mechanisms of sequential neural activities, we focus on a class of low-rank RNNs (*i.e.*, when $W = \sum_{k=1}^{K} m^{(k)} n^{(k)T}$ are constrained to be low-rank). In this class, the computation performed by RNNs can be studied with the latent dynamical system (Dubreuil et al., 2022; Valente et al., 2022; Dinc et al., 2025a; 2023):

$$\tau \dot{\kappa}_k(t) = -\kappa_k(t) + \sum_{i=1}^{N} n_i^{(k)} \times$$
$$\times \tanh\left(\sum_{l=1}^{K} m^{(l)} \kappa_l(t) + W^{\text{in}}u(t) + b\right), \quad (2)$$

where the latent variables are defined as $\kappa_k(t) = n^{(k)T}r(t)$ (Dinc et al., 2025a). Intuitively, by constraining the rank of the weight matrix, $W$, one can project the computation performed by $N$ neurons down to a $K$-dimensional latent dynamical system. In this work, we use this paradigm (*i.e.*, training low-rank RNNs) to illustrate the distinct mechanisms that support sequence generation, as they emerge during training. Then, our large-scale experiments generalize our findings to full-rank RNNs with no explicit restrictions.

## 3. Results

### 3.1. Latent mechanisms underlying neural sequences

As a first step, we set out to answer our first question (*i.e.*, **Q1:** What mechanisms support memory maintenance through sequential neural activity?) using a simple short-term memory task, stripped of any extraneous components, that can be solved using neural sequences (Fig. 1**D-E**). Specifically, we focused on one of the simplest tasks called a delayed activation task (Dinc et al., 2025b). In this task,

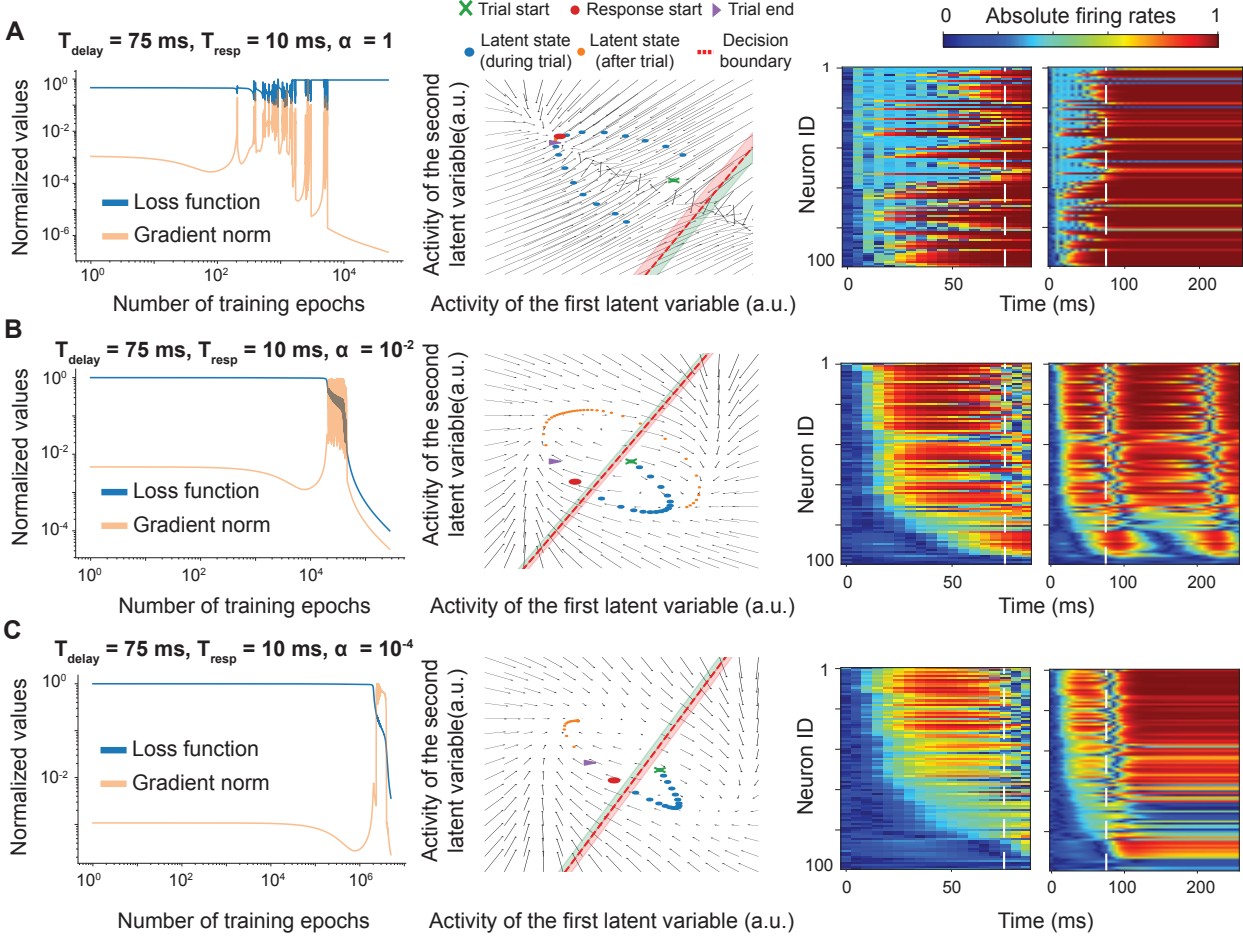

Figure 2. **Changing optimization parameters may result in RNNs learning distinct attractor mechanisms.** We trained a rank-2 RNN to perform the delayed activation task, in which RNNs initialized at a particular state (**Methods**) should delay their output by $T_{\text{delay}}$ for a $T_{\text{resp}}$ time interval. The task does not constrain the network activity or the output outside of these time windows. **A** *Left.* The normalized gradient norm and the loss function values. *Middle.* Flow maps of the learned latent dynamical systems demonstrating the (learned) attractor mechanism. The shaded areas around the decision boundary correspond to network outputs within $0.75$ (outer green boundary) and $0.25$ (outer red boundary). *Right.* Absolute firing rates of the network, whose latent dynamics were illustrated in the middle column. Dotted line corresponds to the start of the response period. Parameters: $N = 100$ neurons, $\tau = 10$ms, $\Delta t = 5$ms, $T_{\text{delay}} = 75$ms, $T_{\text{resp}} = 10$ms, $\alpha = 1$ for stochastic gradient descent using otherwise the default parameters in `Pytorch` (Paszke et al., 2017). **B-C** Same as in panel **(A)**, but trained with learning rates of $10^{-2}$ and $10^{-4}$, respectively.

the network is initialized to a particular state, which is randomly selected in the state space (**Methods**), and has to withhold its response ($\hat{o}(t) = 0$) for a $T_{\text{delay}}$ time window. Afterwards, the network is forced to output a response ($\hat{o}(t) = 1$) for $T_{\text{resp}}$ time window.

*Two distinct mechanisms can subserve neural sequences*–
Broadly, we set out to study two latent mechanisms that may subserve the observed neural sequences. The first mechanism relies on the generation of a set of slow-points, neural states in which the neural dynamics vary slowly ($\dot{r}(t) \approx 0$). In this mechanism, neural sequences emerge during the slow transition of the latent dynamical system through these slow-

points. This transition is often structured, *i.e.*, the same latent trajectory would be followed as the system passes through these regions (Sussillo & Barak, 2013), and therefore can lead to stable and sequential activations of neurons (Fig. 1**B**). The second mechanism relies on the generation of limit cycles, which are defined as closed trajectories that represent sustained, periodic oscillations. In this mechanism, when the oscillation periods are much larger than the trial windows, neural sequences can emerge during the task execution without repetition (Fig. 1**C**). But, once the task concludes, the limit cycles would continue to oscillate and regenerate the sequential activities, whereas the slow-points are transitionary and would not repeat (Fig. 1**B-C**).

Since a minimum of two dimensions is required to generate oscillatory behavior in dynamical systems (Strogatz, 2018), we start by studying the learned latent representations in an RNN with rank $K = 2$ (corresponding to two-dimensional latent dynamics). The output, $\hat{o}(t)$, of the RNN is defined as a nonlinear projection of the latent variables using a sigmoid function $f(\cdot)$. For the largest learning rate (Fig. 2A), this rank-2 RNN was incapable of solving the task despite the nearly-zero gradient[3]. As the learning rates decreased, it was capable of solving the task with *limit cycles* with a moderate learning rate (Fig. 2B), but utilized *slow-points* (also referred to as "SP manifolds" for generality) for the lowest learning rate (Fig. 2C). Notably, both mechanisms produced neural activity sequences during the task window (Fig. 2, right panels).

Despite leading to equivalent neural activities within the trial window, the two mechanisms had qualitatively different behaviors after the trial concluded (Fig. 2). Limit cycles had led to recurrent generations of the same sequences, whereas the SP manifolds converged to a persistent activity style representation after the trial concludes. We will make this observation rigorous in our large-scale studies below. For now, we provide insights into another relevant question: Do RNNs commit to learning one mechanism from the start, or are they capable of changing their inner mechanisms during the training?

*Learned latent mechanisms can evolve during learning*– We next studied the latent dynamical systems of a rank-2 RNN as it was training to solve the delayed activation task (Fig. S1). Interestingly, the loss function showed characteristic periods of learning, corresponding to changes in mechanisms (Fig. S1A). Specifically, after the first jump in the loss function, a SP manifold was formed in the latent dynamical system (Fig. S1B; *left*). Here, a line of slow-points allowed the network output to stall until $T_{\text{delay}}$. Then, the SP manifold had acquired a curved structure (Fig. S1B; *middle*), reminiscing an onset to a rotating solution. Finally, after the oscillations in the loss function values ceased, a limit cycle had emerged in the latent dynamical system (Fig. S1B; *right*). Hence, we confirmed that RNNs could change their inner mechanisms during learning, *i.e.*, what started as a SP manifold solution later evolved into a limit cycle.

In reality, since functional connections in the brain can rapidly reorganize within the start and end of a trial (Ebrahimi et al., 2022), the two mechanisms can become practically inseparable in biological neural networks. In other words, there is an equivalence class of mechanisms that can generate neural sequences that have the same local

---

[3]This phenomenon has been studied in details by a previous work (Dinc et al., 2025b), in which the learning dynamics can get stuck in no-learning zones with near-zero gradients when very large learning rates are used.

behavior within the task window, but can show distinct properties outside; and the two mechanisms we identified belong to this class. For the rest of this work, we will formalize this connection between the learned latent mechanisms and the task and/or optimization parameters. In doing so, we will generalize these findings beyond anecdotal observations via theoretical investigations and large-scale experiments.

### 3.2. Influence of task design on the learned mechanism

Behavioral experiments in systems neuroscience often contain arbitrary design choices that are deeply embedded in the experimental paradigm, yet their implications are rarely scrutinized. Consider the delayed cue discrimination task, a task commonly studied in neuroscience experiments with mice (Ebrahimi et al., 2022). In the machine learning context, it consists of four distinct periods (Fig. 1D-E): cue, delay, reaction, and an optional post-reaction period (i-iv). Upon observation of a cue, the networks have to wait for a brief period and output a response in the corresponding output channel. Typically, there are two possible cues and hence two output channels. If there is a post-reaction period, then the networks should learn to output zeros after the response window concludes. In experimental context, while animals typically indicate their response by licking either a right or left spout, the reward or punishment is often directly provided at the conclusion of the behavior without the post-reaction window. The distinction, as we show below, matters for the final learned solution.

*Minor modifications to the task requirements can change the learned mechanisms*–We now set out to answer our second question (*i.e.*, **Q2:** What determines which mechanism is favored under different task or training conditions?). To do so, we perform a minor change to the delayed activation task by requiring a short post-reaction waiting period and illustrate how minor changes in experimental design can fundamentally alter the mechanisms networks tend to learn during training. In the modified delayed activation task, (rank-2) RNNs are required to switch their responses at the end of the response period and output zero for a brief duration before the trial concludes (Fig. 1D-E). In networks trained without this modification ($T_{\text{post}} = 0ms$), the latent dynamical systems can learn both slow-point manifolds (for shorter delays) and limit-cycles (for longer delays), similar to Fig. 2. However, when we introduce even a brief post-reaction period ($T_{\text{post}} = 30ms$), the networks predominantly learn limit cycle solutions, despite achieving similar task performance (Fig. 3A-B). We quantified and confirmed these findings with a broader empirical study with 1600 RNNs (Fig. 3C; via Fisher's exact test with Bonferroni corrections). Thus, in rank-2 RNNs solving the delayed activation task, increasing the delay period promoted the emergence of limit cycle solutions, and this tendency was dramatically amplified by the addition of a post-reaction

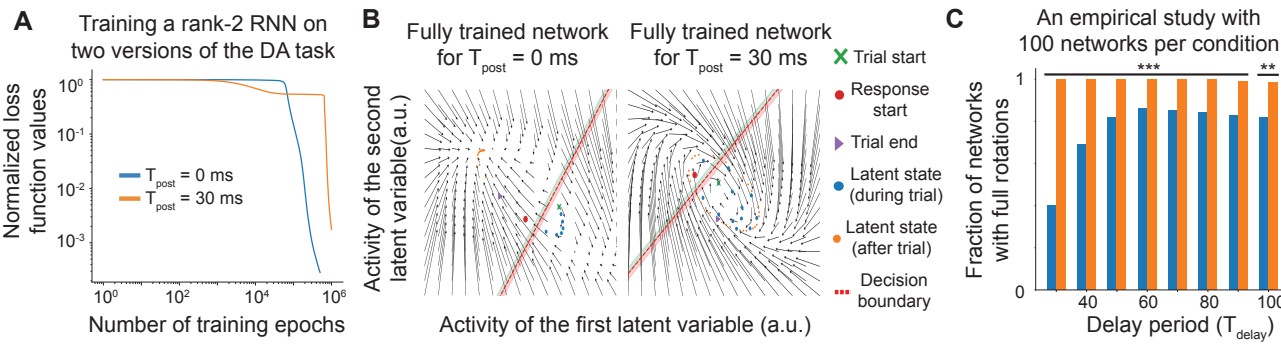

*Figure 3.* **Changing trivial task parameters may result in RNNs favoring limit cycle solutions. A** We performed experiments similar to those in Fig. 2, but now with an added post-reaction period ($T_{\text{post}}$). **B** Learned latent dynamics for the fully trained networks in panel (**A**). Parameters for **A-B**: As in Fig. 2A, with $T_{\text{delay}} = 30\text{ms}$, $T_{\text{resp}} = 10\text{ms}$, $T_{\text{post}} = 0$ or $30\text{ms}$, and $\alpha = 10^{-3}$. **C** Fraction of networks developing rotating solutions (*i.e.*, limit cycles) when solving the two versions of the task as a function of delay $T_{\text{delay}}$. Parameters: $\alpha = 0.005$, varying levels training epochs corresponding to $T_{\text{delay}}$, otherwise the same as in panels **A-B**. Comparisons: Fisher's exact test with Bonferroni corrections, ***$p < 0.001$, and **$p < 0.01$.

window. This demonstrated that seemingly minor changes in the task design can favor learning distinct short-term memory maintenance mechanisms.

### 3.3. A theoretical scaling analysis of mechanistic phases

So far, we used simple examples to demonstrate how RNNs can use different latent mechanisms to generate neural sequences, that RNNs can change their inner mechanisms during the course of their training, that the choice between these mechanisms is not random but depends on specifics of the experimental design. Now, to provide an answer to the third question (*i.e.*, **Q3:** How do the learned mechanisms vary with delay duration?), we turn to a set of low-dimensional dynamical models and study their learning dynamics.

*Why study low-dimensional toy models?*–Recent work has shown that training (full-rank) RNNs often converges to effectively low-dimensional solutions (Valente et al., 2022), a property that appears to generalize across many real-world networks (Thibeault et al., 2024). As discussed in Eq. (2), these low-dimensional computations can often be interpreted as latent dynamical systems. In this view, training the RNN parameters $(W, W^{\text{in}}, b)$ amounts to shaping a latent flow map of the form:

$$\tau\dot{\kappa}(t) = G(\kappa(t), u(t); W, W^{\text{in}}, b), \qquad (3)$$

where $G$ governs the evolution of low-dimensional latent variables $\kappa(t)$ in response to input $u(t)$. Hence, studying how toy models, which do not necessarily share the same architectural assumptions as RNNs but still aim to learn similar local dynamics, may provide intuition for how these flow maps may be learned. Thus, we set out to analyze simple low-dimensional dynamical systems that capture the two mechanisms (slow-points and limit cycles) we have observed in trained RNNs.

Earlier work has considered a one-dimensional toy system, the normal form of a saddle-node bifurcation, $\dot{x} = x^2 + r$ (Strogatz, 2018), to study the learning dynamics of slow-point formation (Dinc et al., 2025b). This system approximates the local behavior of one-dimensional flow maps $G(\kappa, u)$ near their minima, where slow-points (also observed in Fig. 2) arise. To perform a similar analysis with the oscillatory dynamics observed in Figs. 2 and 3, we will consider a second toy model in this work: a two-dimensional system exhibiting a limit cycle, one of the simplest settings in which periodic orbits, and hence oscillations, can emerge (Strogatz, 2018). Both toy models are meant to qualitatively mirror the behavior of the respective latent flow maps $G(\kappa, u)$ that may be learned by an RNN (Fig. 2). While these toy systems are not exhaustive, they capture key features of the latent mechanisms that RNNs may converge to, and allow us to dissect their learning dynamics in a controlled and *analytical* setting, which we discuss next.

*Extracting scaling laws from toy models*–To start with, in Appendix S2.1, we reproduce the findings of the earlier work on slow-point manifolds for a more general version of the delayed activation task with $T_{\text{delay}} \neq T_{\text{resp}}$. This analysis reveals a maximum learning rate, also reported in (Dinc et al., 2025b), beyond which the learning dynamics destabilize and even an approximation of the desirable solution cannot be reached. Our analysis reveals that this maximum learning rate follows a power law such that $\alpha_{\text{SP}} \sim O(T_{\text{delay}}^{-\beta_{\text{SP}}})$, where $\beta_{\text{SP}} \in [4, 5]$. The exact value $\beta_{\text{SP}}$ achieves within this interval depends whether $T_{\text{delay}}$ and $T_{\text{resp}}$ are comparable, with larger delays leading to more substantial decreases (approaching $\beta_{\text{SP}} \to 5$ in the limit $T_{\text{delay}} \gg T_{\text{resp}}$). Such a steep decrease with the desirable learning rate can explain our anecdotal observations in Figs. 2 and 3**C** that slow-points become rarer for large $T_{\text{delay}}$ or high learning rates, which will be substantiated in the next section.

Now, we focus on the limit cycle model. For simplicity, we fix the radius to an attractive fixed-point, and allow a periodic oscillation in the frequency such that the equations of the motion for the toy model become (See Appendix S2.2 for derivations) $x(t) = \sin(2\pi rt)$, where $r \in \mathbb{R}$ is a learnable parameter. The network output is defined as $\hat{o}(x) = \Theta(x < 0)$, where $\Theta(\cdot)$ is defined as the Heaviside function. By inspection, the rotations have a half-period of $T_{\text{half}} = \frac{1}{2r}$, and the model outputs zero for the first $T_{\text{half}}$ while outputting one for the second half. An example flow map is shown in Fig. S2**A**; changing the parameter, $r$, modulates the frequency/period of the cycle.

In Appendix S2.2, we compute an analytical loss function for the limit cycle model in the limits $T_{\text{resp}} = T_{\text{delay}}$ and $T_{\text{resp}} \ll T_{\text{delay}}$. We illustrate the latter case in Fig. S2**B**, note the exact correspondence between the analytical curve and the numerical simulations. For both cases, the loss function has a global optimum at $r = \frac{1}{2T_{\text{delay}}}$. However, it has a kink around the global minimum and quickly flattens to a constant value for $r \leq \frac{1}{2(T_{\text{delay}} + T_{\text{resp}})}$. Hence, if the learning rate is larger than some critical value, the oscillations around the global minimum can throw the parameter $r$ to a value $r < \frac{1}{2(T_{\text{delay}} + T_{\text{resp}})}$, terminating the training process due to the zero gradient despite the fact that the model cannot solve the task. Our analysis reveals that this maximum learning rate scales following $\alpha_{\text{LC}}^* \sim T_{\text{delay}}^{-\beta_{\text{LC}}}$ where $\beta_{\text{LC}} \in [2, 3]$. Notably, given that $\beta_{\text{LC}} \leq \beta_{\text{SP}}$, the limit cycle solution has a better scaling than the slow-point one, providing a theoretical explanation for our observation in Fig. 2. Specifically, since the RNN in Fig. 2 was trained using a learning rate right at the transition (compare to Fig. S1), it is not surprising that a slow point mechanism eventually evolved into a limit cycle. Moreover, this theoretical prediction is also consistent with the experiments in Fig. 3**C**, where increasing the delay period for a fixed learning rate resulted in more RNNs finding limit cycle based solutions. Hence, both learning rate and the duration of the delay can influence the learned latent mechanism during training.

*Further insights from toy models*– The distinct scaling of $\beta_{\text{LC}}$ and $\beta_{\text{SP}}$ also highlights the existence of an interesting phenomenon: a phase space of mechanisms that can be learned by RNNs. This phase space (which will be discussed in Fig. 4**G**) depends on the delay time (a task parameter) and the learning rate (an optimization parameter). On the other hand, it is important to not over-interpret the toy model results. Specifically, increasing the dimensionality of the latent dynamical system can allow more efficient solutions to the task, and it may be possible for RNNs to utilize other (diverse) mechanisms for storing memories (*e.g.*, those using inherent transient dynamics in high-dimensional neural networks (Jaeger, 2002; Liu et al., 2025; Maass et al., 2002; Ichikawa & Kaneko, 2021)). Fortunately, the toy models we discussed utilize mechanisms that are observed in practice

(Figs. 1, 2, and S1), and therefore can offer further insights into learning dynamics in more complicated architectures. Below, we provide one example case, and show empirically that similar phase spaces emerge in full-rank RNNs trained on delayed cue discrimination tasks.

### 3.4. Empirical scaling analyses of latent mechanisms

As a final step, we tested the theoretical predictions of the toy models, including the proposed scaling laws and the phase space of latent mechanisms, by training full-rank RNNs on delayed cue-discrimination tasks (Figs. 4, S3, S4, and S5). Similar to low-rank RNNs, full-rank RNNs also learned to solve the task using two distinct mechanisms (Fig. 4**A**; Fig. S3**A,B**). Solutions involving sustained oscillations exhibited minimal power at low frequencies, as the dynamics were dominated by high-frequency components (Fig. S3**C**). Based on this observation, we developed a mechanism discrimination index that automatically classified the learned RNNs into three groups: (i) networks relying on slow-point manifolds, (ii) networks employing limit cycles, and (iii) networks that failed to perform the task (see Fig. S4 for representative examples).

*Large-scale RNN experiments reproduced the toy model predictions*–We then applied this classification to examine whether different mechanisms consistently emerged across varying delay durations. In line with theoretical predictions, increasing the length of the delay period preferentially led to the emergence of limit cycles (Fig. 4**B**). These oscillatory solutions disappeared beyond a critical learning rate for a given delay, with the transition scaling as $\beta_{\text{LC}} = 2.72 \pm 0.07$ (Fig. 4**C**), in close agreement with predictions (see Fig. S3**D-E** for the procedure used to compute critical delays). For shorter delays, RNNs predominantly developed slow-point manifolds, with the transition scaling as $\beta_{\text{SP}} = 4.05 \pm 0.10$, also matching the theoretical range (Fig. S3). However, when we trained RNNs on a modified task variant that included a post-reaction window, slow-point solutions were no longer observed, and only oscillatory dynamics emerged (Fig. 4**D-F**). To summarize these transitions, we plotted phase diagrams (Fig. 4**G-I**) derived from both the theoretical model and the full-rank RNN experiments, capturing the conditions under which each mechanism emerged and their associated scaling laws.

*Control experiments*–Finally, we performed two additional control experiments. In Fig. S5, we repeated the analysis using RNNs with slower intrinsic dynamics (i.e., larger $\tau$), and in Fig. S6, we tested a version of the task that no longer required memory maintenance. In the latter case, the dependence on task duration nearly vanished, and the scaling observed in Fig. 4 was substantially weakened ($\beta = 0.38 \pm 0.02$). These findings reinforce the idea that the computational mechanisms adopted by RNNs are not fixed

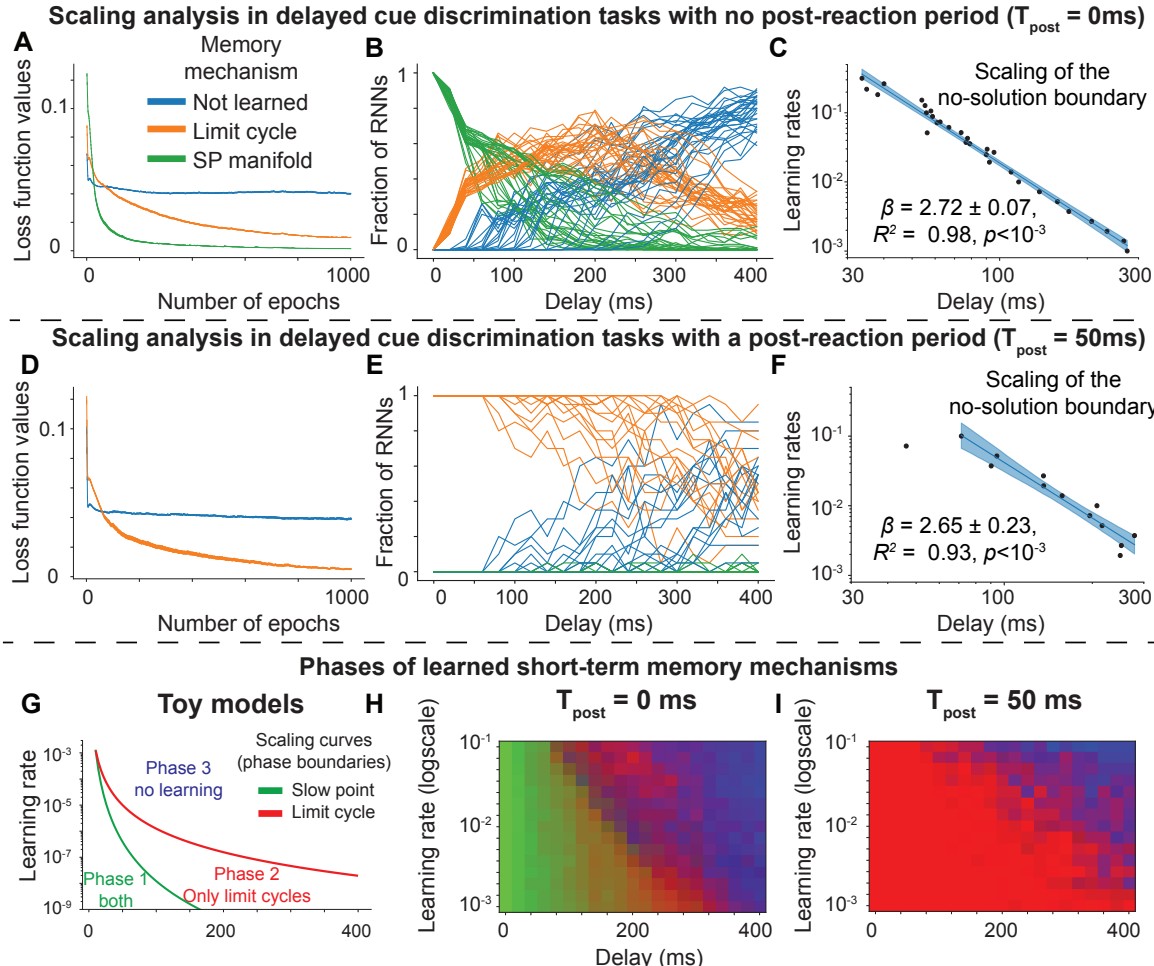

*Figure 4.* **Emergent phases of attractor mechanisms in RNNs performing delayed cue-discrimination. A** Loss curves corresponding to different memory mechanisms observed during training. **B** Distributions of learning rates and resulting memory mechanisms as a function of the delay period. Each line represents a distinct learning rate; each run is initialized with a different random seed. **C** Power-law fits shown on a log-log scale, where the $y$-axis indicates the maximum learnable delay and the $x$-axis the corresponding critical learning rate. Each dot denotes the cutoff delay value beyond which learning fails (see **Methods** for how these values are estimated). **D-F** Same as in panels **A-C**, but for tasks with post-reaction windows. **G** The phase diagram of mechanisms predicted by the toy model. **H-I** Phase diagrams corresponding to panels **B-E**, where color intensity reflects the proportion of trials converging to each of the three dynamical phases. As predicted by the theoretical analysis in **G**, the original task (**H**) reveals three distinct regimes, whereas the inclusion of a post-reaction window (**I**) biases the model toward limit cycle dynamics. To generate these phase diagrams, we trained over 65,000 RNNs across varying learning rates, delay intervals ($T_{\text{delay}}$), and random seeds (see Table S1). For full parameter details, refer to Section S4.

but shaped by the task's demands. In particular, memory duration, learning rate, and structural constraints (refer to **Q3** above) jointly determine the preferred mechanism.

## 4. Discussion

In this work, we set out to answer three fundamental questions about the population-level mechanisms that can support short-term memory in neural networks.

*What mechanisms support memory maintenance through sequential neural activity?* In the memory tasks studied here, neural activation sequences consistently emerged as

core components of the solutions. These sequences arose through two distinct mechanisms: *slow-point manifolds* and *limit cycles*. While the use of slow-points to approximate sequences is well aligned with prior intuitions (Khona & Fiete, 2022), the emergence of limit cycles in this context represents a novel prediction of our framework. Although limit cycles have previously been observed in memory tasks (Pals et al., 2024), they typically appear when periodicity is embedded in the task design. In contrast, the delayed activation and delayed cue-discrimination tasks used here had no explicit periodic components. Thus, the appearance of limit cycles in these tasks suggests an alternative, previously un-

derappreciated, mechanism for generating sustained neural sequences in the absence of external rhythmic input or an oscillatory output.

*What determines which mechanism is favored under different memory tasks or training conditions?* Our analytical results revealed that the selection between slow-point manifolds and limit cycles depended on both the structure of the task and the learning rate used in the optimization. We later confirmed these predictions in large-scale simulations using full-rank RNNs (Table S1). This finding is particularly striking in light of recent interest in dynamical systems approaches in neuroscience, where considerable effort has gone into identifying and manipulating attractors and manifolds that underlie neural activity (Liu et al., 2024; Vinograd et al., 2024). The sharp divergence in dynamical solutions (slow-points versus limit cycles) emerging from minor changes in task design underscores a critical point: the structure of the task can strongly shape the neural dynamics that emerge.

This raises important questions about how we interpret dynamical structures observed in neural recordings. Can we conclude that one type of solution is more fundamental than another simply because it appears in a specific task variant? Our findings highlight the need for caution. They suggest that conclusions about the nature of neural computations must be grounded not only in observed dynamics but also in a careful analysis of how task parameters may constrain or bias the underlying solutions. What may appear to be a core computational motif could, in fact, reflect a contingent outcome of the task design itself.

*How do the learned mechanisms vary with the delay duration?* A key insight into this question comes from examining how critical learning rates (beyond which learning ceases to be effective) scale with delay length. In simple delayed-response tasks, we found that the critical learning rate allowing successful training decreases as a power-law function of the delay. As the delay increases, smaller and smaller updates are required, ultimately making training impractically slow for long durations. This constraint may underlie the historical difficulty of capturing long-term dependencies in RNNs, as highlighted in early work (Bengio et al., 1994).

The introduction of gated architectures such as LSTMs and GRUs partially addressed this issue by enabling networks to learn adaptive timescales (Hochreiter, 1997). These architectures could perhaps bypass the power-law constraint by dynamically modulating their internal time-scales, and thereby extend their memory retention periods. However, even if successful, this flexibility would lead to trade-offs. Extending timescales indefinitely slows down network responses, limiting their utility for rapid decision-making. In addition, expanding the parameter space to accommo-

date longer memory spans can complicate optimization and destabilize learning (Dinc et al., 2025b). Unlike biological systems, which operate under biophysical constraints such as membrane time constants and action potentials, artificial networks must learn timescales through training, which is not always possible even if the underlying network is theoretically capable of solving a particular task.

Overall, by framing the challenge through dynamical systems theory, our work highlights a core trade-off between memory duration, learning speed, and network stability. Understanding this trade-off offers a principled explanation for the persistent difficulty of learning long-term dependencies and opens the door to new strategies that balance biological plausibility, computational efficiency, and learning performance.

*Concluding remarks*—Our findings carry important implications for experimental neuroscience. We have shown that both slow-points and limit cycles can serve as effective mechanisms for generating sequential neural activity. However, in the context of a dynamic brain, where synaptic plasticity continually reshapes functional connectivity (Ebrahimi et al., 2022), it remains unclear how to determine whether observed sequences are supported by slow-points or limit cycles. More broadly, how can we distinguish between dynamical solutions that produce equivalent activity patterns during a trial, but diverge in their behavior once the trial ends? To our knowledge, this question (concerning equivalence of distinct latent mechanisms subserving task execution, but later diverging after the task conclusion) has not been explicitly addressed in experimental designs within systems neuroscience. Tackling it would require a shift in how experiments are conceptualized and conducted. As a starting point, our work illustrates how (even minor) manipulations of task structure can lead to distinct underlying mechanisms.

Beyond neuroscience, the phenomenon of staged learning dynamics and algorithmic phase transitions has been widely documented in artificial neural networks, including Transformers trained on modular arithmetic (Liu et al., 2022; Nanda et al., 2023), language modeling (Wei et al., 2022; Lubana et al., 2024), and in-context learning (Raventós et al., 2024). Within this growing literature, our study of RNNs provides a complementary perspective, grounded in dynamical systems theory, that connects computational behavior to interpretable latent mechanisms. Recent efforts aim to formalize learning progress in feedforward networks using quantities analogous to order parameters in physics (Park et al., 2024). Our results go a step further by identifying literal phase transitions between qualitatively distinct computational strategies in dynamical systems. Future work may explore how these mechanisms can be implemented or approximated by feedforward architectures.

## Acknowledgements

We would like to thank members of the Geometric Intelligence Lab for their helpful feedback on an earlier version of the project and Dr. Boris Shraiman for fruitful discussions on the final manuscript. EC and BK's internships were supported in part by a grant from the Feldman-McClelland Open-a-Door fund of the Pittsburgh Foundation. BK thanks the Impact Scholarship and Research Scholarship for Critical Thinkers programs from the Bridge to Turkiye Funds for supporting his visit to Stanford University. FD expresses gratitude for the valuable mentorship he received at PHI Lab during his internship at NTT Research and acknowledges funding from Stanford University's Mind, Brain, Computation and Technology program, which is supported by the Stanford Wu Tsai Neuroscience Institute.. MJS gratefully acknowledges funding from the Simons Collaboration on the Global Brain and the Vannevar Bush Faculty Fellowship Program of the U.S. Department of Defense. NM acknowledges funding from the National Science Foundation, award 2313150. PY was additionally supported by the Foundation of the Shanghai Municipal Education Commission (No. 24RGZNA01). This research was supported in part by grant NSF PHY-2309135 and the Gordon and Betty Moore Foundation Grant No. 2919.02 to the Kavli Institute for Theoretical Physics (KITP). Some of the computing for this project was performed on the Sherlock cluster. We would like to thank Stanford University and the Stanford Research Computing Center for providing computational resources and support that contributed to these research results.

## Impact Statement

This paper contributes to machine learning and computational neuroscience by making an unprecedented number of trained RNNs publicly available—models that required several weeks of compute time to generate.

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

# Methods

## S1. Theoretical details on attractor mechanisms subserving short-term memory

### S1.1. A short background on dynamical system theory

A dynamical system describes how the state of a system evolves over time. Formally, it can be represented by a differential equation of the form:

$$\dot{x}(t) = f(x(t)), \quad x(t_0) = x_0,$$

where $x_0$ denotes the initial state, and $\dot{x}$ represents the time derivative of the state variable $x \in \mathbb{R}^n$.

In neuroscience, researchers often examine neural firing rates $r \in \mathbb{R}^n$ through the lens of dynamical systems. This approach helps uncover how temporal dynamics unfold within lower-dimensional neural state spaces, as characterized by dynamical systems theory.

Since firing rates are themselves components of the system's state, in this study we simulate such systems using leaky firing-rate RNNs—biologically interpretable neural networks. One might argue that RNNs may not be expressive enough to learn such complex dynamics and that more sophisticated architectures like Long Short-Term Memory networks (LSTMs) or Transformers might perform better. However, this concern is addressed by the universal approximation property of RNNs, which states that, given a sufficient number of neurons, even vanilla RNNs can approximate any dynamical system (Schäfer & Zimmermann, 2006). Additionally, compared to more complex models, RNNs offer greater interpretability due to their simpler and shallower architecture.

As previously mentioned, neuroscientists often study neural activity through the lens of attractor dynamics. In this framework, researchers analyze how systems evolve over time by characterizing their underlying attractor structures. As we focus on several commonly studied attractors from dynamical systems theory (Strogatz, 2018), we describe the fixed-point attractors, slow-point manifolds, and limit cycles below.

**Fixed-Point Attractors:** In a fixed-point attractor, the system evolves toward a specific state where it remains indefinitely. Once the state is reached, the time derivative becomes zero, i.e., $\dot{x} = 0$.

**Slow-Point Manifolds:** Slow-point manifolds are particularly relevant to memory-related mechanisms in neuroscience. These manifolds describe regions of the state space where the system changes very slowly, satisfying $\dot{x} \approx 0$. Unlike fixed points, which trap the system, slow points allow it to persist along a trajectory over time. This property enables the system to maintain information temporarily, as illustrated in Fig. S4**D**.

**Limit Cycle Attractors:** Limit cycle attractors describe systems that exhibit sustained oscillatory behavior. These trajectories form closed loops in state space and are often associated with rhythmic or repeated tasks. In such cases, $\dot{x} \neq 0$, as shown in Fig. S4**C**. However, as demonstrated in our experiments, limit cycles can closely approximate slow-point manifolds during the trial period.

Importantly, a single dynamical system may contain multiple attractor types, and the specific trajectory it follows can depend on the initial condition $x_0$. Nonetheless, identifying and understanding these fundamental attractor structures is essential for uncovering the mechanisms underlying neural population dynamics.

## S2. Theoretical studies of scaling relationships via toy models

In this section, we provide the additional details and calculations regarding the toy models we introduced in the main text.

### S2.1. The ghost model

The first model we considered is the ghost model, which effectively forms a slow point that can be used to store the duration of the delay before the network takes an action. This model was introduced in a prior work (Dinc et al., 2025b), whose results we briefly summarize below. Then, we generalize the methodology introduced in the earlier work, which we will eventually use to study the additional toy models we introduced in this work.

The toy model starts with a one-dimensional dynamical system:

$$\tau \dot{x}(t) = x^2(t) + r, \tag{S1}$$

where $x \in \mathbb{R}$ is the state variable and $r \in \mathbb{R}$ is a learnable parameter. Here, $\tau$ is a fixed time scale, which we effectively set to one such that time is computed in dimensionless units (in units of $\tau$). This system is very well known in the traditional dynamical system theory literature as the standard form for the saddle-node bifurcation (Strogatz, 2018). Specifically, for $r < 0$, the dynamical system has two fixed points $x^* = \pm\sqrt{-r}$, the one on the left being attractive and the other repulsive. However, for $r > 0$, the system has no fixed-points. Hence, a "saddle-node bifurcation" is said to take place as $r$ is varied between positive and negative values (Strogatz, 2018). Since local minima of most functions can be approximated by a quadratic term, this is considered the standard form for the saddle-node bifurcation and can be used to approximate the emergence/collusion of two fixed-points.

Though traditional work often focuses on changes as $r$ is varied by hand, (Dinc et al., 2025b) has taken an alternative approach and considered learning this parameter through a task. Specifically, the output of the network is defined as $\hat{o}(x) = \Theta(x - x^*)$, where $\Theta(\cdot)$ is the Heaviside function and $x^*$ is some fixed-parameter (which is later taken to $\infty$ in analytical calculations). The model is initialized at $x(t = 0) = 0$. Then, the goal of the model is to output 0 until some time $T$ and then 1 for another $T$ times. In mathematical terms, $o(t) = \Theta(t - T)$. Unlike the prior work (Dinc et al., 2025b), since our goal is to understand the scaling of the effective learning rates with respect to the delay period lengths, we define a *normalized* loss function as:

$$\mathcal{L}(r) = \frac{1}{T} \int_0^{2T} (\hat{o}(x(t)) - o(t))^2 \, dt. \tag{S2}$$

It is possible to compute this loss function exactly in the limit $x^* \to \infty$, which is as follows (Dinc et al., 2025b):

$$\mathcal{L}(r) = \begin{cases} \left| 1 - \frac{\pi}{2T\sqrt{r}} \right| & \text{for} \quad r \geq \frac{r^*}{4}, \\ 1 & \text{otherwise.} \end{cases} \tag{S3}$$

This loss function achieves the global minimum at $r^* = \frac{\pi^2}{4T^2}$ such that $\mathcal{L}(r^*) = 0$. One can compute the analytical gradient as:

$$\nabla\mathcal{L}(r) = \begin{cases} -\frac{\pi}{4Tr^{3/2}} & \text{for } \frac{\pi^2}{16T^2} < r < \frac{\pi^2}{4T^2}, \\ \frac{\pi}{4Tr^{3/2}} & \text{for } r > \frac{\pi^2}{4T^2}, \\ 0 & \text{for } r < \frac{\pi^2}{16T^2}. \end{cases} \tag{S4}$$

Here, the zero gradient for $r < \frac{\pi^2}{16T^2}$ is practically undesirable, since the loss function is still quite high. This flat region, also observed in other toy models of our interest, is referred to as a "no learning zone." Once a model is caught in this region, there is no returning back.

Moreover, in this toy model, an interesting phenomenon emerges around the global minimum. Specifically, the derivative at $r = r^* = \frac{\pi^2}{4T^2}$ becomes:

$$\text{At } r = \frac{\pi^2}{4T^2}: \quad \nabla\mathcal{L}(r) = \begin{cases} -\frac{2T^2}{\pi^2} & \text{from the left,} \\ \frac{2T^2}{\pi^2} & \text{from the right.} \end{cases} \tag{S5}$$

These expressions differ from (Dinc et al., 2025b) only in the sense that the normalization of the loss function introduces a $(T)^{-1}$ coefficient to the gradient computation. Notably, this loss function has a kink at its global optimum. Thanks to this kink, (Dinc et al., 2025b) has defined a maximum (effective) learning rate $\alpha^*$, for any $\alpha > \alpha^*$, the model would be thrown from the global minimum to the zero gradient regime, after which the learning would come to an halt despite a non-desirable loss value of 1. This learning rate can be computed as:

$$\alpha^*_{\text{ghost}} \left. |\nabla\mathcal{L}(r)| \right|_{r \to r^*_+} = \frac{3\pi^2}{16T^2} \implies \alpha^*_{\text{ghost}} = \frac{3\pi^4}{32} T^{-4}. \tag{S6}$$

Thus, for a normalized loss function, the effective learning rate for the ghost point toy model scales following $O(T^{-4})$. Below, we extend this toy model with asymmetric response ($T_{\text{resp}}$) and delay ($T_{\text{delay}}$) times.

S2.1.2. GHOST MODEL WITH ASYMMETRIC RESPONSE TIME

For the problem of our interest, we simply replace the loss function with:

$$\mathcal{L}(r) = \frac{1}{T_{\text{delay}}} \int_0^{T_{\text{delay}}} (\hat{o}(x(t)) - 0)^2 \, dt + \frac{1}{T_{\text{resp}}} \int_{T_{\text{delay}}}^{T_{\text{delay}}+T_{\text{resp}}} (\hat{o}(x(t)) - 1)^2 \, dt,$$

$$= \frac{1}{T_{\text{delay}}} \int_0^{T_{\text{delay}}} \hat{o}(x(t)) \, dt + \frac{1}{T_{\text{resp}}} \int_{T_{\text{delay}}}^{T_{\text{delay}}+T_{\text{resp}}} [1 - \hat{o}(x(t))] \, dt, \tag{S7}$$

As shown by previous work (Dinc et al., 2025b) (by integrating the dynamical system equations), in the limit $x^* \to \infty$, the network output can be written as $\hat{o}(t) = \Theta(t - t^*)$, where $t^* = \frac{\pi}{2\sqrt{r}}$. This creates three distinct regimes of analytical calculation: i) $t^* \leq T_{\text{delay}}$, ii) $T_{\text{delay}} \leq t^* \leq T_{\text{delay}} + T_{\text{resp}}$, and iii) $T_{\text{delay}} + T_{\text{resp}} \leq t^*$.

Let us start with the first case, $t^* \leq T_{\text{delay}}$ (*i.e.*, $r \geq \frac{\pi^2}{4T_{\text{delay}}^2}$):

$$\mathcal{L}(r) = \underbrace{\frac{1}{T_{\text{delay}}} \int_0^{t^*} \hat{o}(x(t)) \, dt}_{=0} + \frac{1}{T_{\text{delay}}} \int_{t^*}^{T_{\text{delay}}} \hat{o}(x(t)) \, dt + \underbrace{\frac{1}{T_{\text{resp}}} \int_{T_{\text{delay}}}^{T_{\text{delay}}+T_{\text{resp}}} [1 - \hat{o}(x(t))] \, dt}_{=0},$$

$$= \frac{1}{T_{\text{delay}}} \int_{t^*}^{T_{\text{delay}}} dt = 1 - \frac{t^*}{T_{\text{delay}}} = 1 - \frac{\pi}{2T_{\text{delay}}\sqrt{r}}. \tag{S8}$$

Next, we consider the second case, $T_{\text{delay}} \leq t^* \leq T_{\text{delay}} + T_{\text{resp}}$ (*i.e.*, $\frac{\pi^2}{4(T_{\text{delay}}+T_{\text{resp}})^2} \leq r \leq \frac{\pi^2}{4T_{\text{delay}}^2}$):

$$\mathcal{L}(r) = \underbrace{\frac{1}{T_{\text{delay}}} \int_0^{T_{\text{delay}}} \hat{o}(x(t)) \, dt}_{=0} + \frac{1}{T_{\text{resp}}} \int_{T_{\text{delay}}}^{t^*} [1 - \hat{o}(x(t))] \, dt + \underbrace{\frac{1}{T_{\text{resp}}} \int_{t^*}^{T_{\text{delay}}+T_{\text{resp}}} [1 - \hat{o}(x(t))] \, dt}_{=0},$$

$$= \frac{1}{T_{\text{resp}}} \int_{T_{\text{delay}}}^{t^*} dt = \frac{t^*}{T_{\text{resp}}} - \frac{T_{\text{delay}}}{T_{\text{resp}}} = \frac{\pi}{2T_{\text{resp}}\sqrt{r}} - \frac{T_{\text{delay}}}{T_{\text{resp}}}. \tag{S9}$$

Finally, we consider the third case, $T_{\text{delay}} + T_{\text{resp}} \leq t^*$ (*i.e.*, $r \leq \frac{\pi^2}{4(T_{\text{delay}}+T_{\text{resp}})^2}$):

$$\mathcal{L}(r) = \underbrace{\frac{1}{T_{\text{delay}}} \int_0^{T_{\text{delay}}} \hat{o}(x(t)) \, dt}_{=0} + \frac{1}{T_{\text{resp}}} \int_{T_{\text{delay}}}^{T_{\text{delay}}+T_{\text{resp}}} [1 - \hat{o}(x(t))] \, dt$$

$$= \frac{1}{T_{\text{resp}}} \int_{T_{\text{delay}}}^{T_{\text{delay}}+T_{\text{resp}}} dt = 1. \tag{S10}$$

Bringing all these cases together, the loss function becomes:

$$\mathcal{L}(r) = \begin{cases} 1 & \text{for} \quad r \leq \frac{\pi^2}{4(T_{\text{delay}}+T_{\text{resp}})^2}, \\ \frac{\pi}{2T_{\text{resp}}\sqrt{r}} - \frac{T_{\text{delay}}}{T_{\text{resp}}} & \text{for} \quad \frac{\pi^2}{4(T_{\text{delay}}+T_{\text{resp}})^2} \leq r \leq \frac{\pi^2}{4T_{\text{delay}}^2}, \\ 1 - \frac{\pi}{2T_{\text{delay}}\sqrt{r}} & \text{for} \quad r \geq \frac{\pi^2}{4T_{\text{delay}}^2}. \end{cases} \tag{S11}$$

Firstly, we note that this loss function is continuous and achieves its global minimum at $r^* = \frac{\pi^2}{4T_{\text{delay}}^2}$ such that $\mathcal{L}(r^*) = 0$. We can once again compute the derivative near the global minimum:

$$\text{At } r = \frac{\pi^2}{4T_{\text{delay}}^2} : \quad \nabla \mathcal{L}(r) = \begin{cases} -\frac{2T_{\text{delay}}^3}{T_{\text{resp}}\pi^2} & \text{from the left,} \\ \frac{2T_{\text{delay}}^2}{\pi^2} & \text{from the right.} \end{cases} \tag{S12}$$

Similar to before, one can compute the critical learning rate due to the kink at the global optimum. First, noting $r_b = \frac{\pi^2}{4(T_{\text{delay}}+T_{\text{resp}})^2}$, we arrive at:

$$r^* - r_b = \frac{\pi^2}{4T_{\text{delay}}^2} - \frac{\pi^2}{4(T_{\text{delay}}+T_{\text{resp}})^2} = \frac{\pi^2(2T_{\text{delay}}+T_{\text{resp}})T_{\text{resp}}}{4T_{\text{delay}}^2(T_{\text{delay}}+T_{\text{resp}})^2} \xrightarrow{T_{\text{delay}}\gg T_{\text{resp}}} \frac{\pi^2 T_{\text{resp}}}{2T_{\text{delay}}^3}. \tag{S13}$$

Then, the critical learning rate can be found as:

$$\alpha_{\text{ghost}-\text{as}}^* = \frac{r - r_b}{|\nabla \mathcal{L}(r_+^*)|} \approx \frac{\pi^2 T_{\text{resp}}}{2T_{\text{delay}}^3} \frac{\pi^2}{2T_{\text{delay}}^2} = \frac{\pi^4 T_{\text{resp}}}{4} T_{\text{delay}}^{-5}. \tag{S14}$$

In words, if the delay period is significantly longer than the response period, the critical learning rate scales as $\sim O(T_{\text{delay}}^{-5})$.

## S2.2. Limit cycle model

Similar to a ghost model, another potential strategy for a delayed response involves creating limit cycle attractors. Though there are multiple dynamical system forms one can use to model these attractors, we focus on a simple toy model as follows:

$$\tau \dot{\rho}(t) = (1 - \rho^2(t))\rho(t), \quad \tau \dot{\theta}(t) = 2\pi r, \tag{S15}$$

where $\rho$ and $\theta$ are polar coordinates and $r$ is the trainable parameter as before. Here, $\tau$ is a fixed time scale, which we effectively set to one such that time is computed in dimensionless units (in units of $\tau$). We assume that the system is initialized at $\rho = 1$ and $\theta = -\pi/2$, and primarily focus on $x(t) = \rho(t)\cos(\theta(t))$. Below, we discuss how we train this model to solve the delayed response task.

### S2.2.1. TOY MODEL SETUP

Since $\rho = 1$ is a fixed point of the limit cycle system with $\dot{\rho}(t)|_{\rho=1} = 0$ and $\theta(t) = 2\pi rt - \frac{\pi}{2}$, the time evolution of the $x$-coordinate follows:

$$x(t) = \sin(2\pi rt), \tag{S16}$$

where $r$ is a learnable parameter. Next, define the output of this system as $\hat{o}(x(t)) = \Theta(x(t) < 0)$. For the delayed response task, we make a slight modification and assume that the delay is longer than the reaction time such that $o(t) = \Theta(T - T_{\text{delay}})$ and the total task time is $T_{\text{total}} = T_{\text{delay}} + T_{\text{resp}}$. Then, the loss function becomes:

$$\begin{aligned}
\mathcal{L}(r) &= \frac{1}{T_{\text{delay}}} \int_0^{T_{\text{delay}}} (\Theta(x(t) < 0) - 0)^2 \, dt + \frac{1}{T_{\text{resp}}} \int_{T_{\text{delay}}}^{T_{\text{delay}}+T_{\text{resp}}} (\Theta(x(t) < 0) - 1)^2 \, dt, \\
&= \frac{1}{T_{\text{delay}}} \int_0^{T_{\text{delay}}} \mathbb{1}(\sin(2\pi rt) < 0) \, dt + \frac{1}{T_{\text{resp}}} \int_{T_{\text{delay}}}^{T_{\text{delay}}+T_{\text{resp}}} \mathbb{1}(\sin(2\pi rt) > 0) \, dt
\end{aligned} \tag{S17}$$

When $T_{\text{resp}} \leq T_{\text{delay}}$, a simple inspection reveals that the global optimum of this problem is achieved when $T_{\text{delay}}$ corresponds to the half the period, *i.e.*, $r^* = \frac{1}{2T_{\text{delay}}}$. At this value, the network outputs 0 until $T_{\text{delay}}$, and 1 for another $T_{\text{delay}} \geq T_{\text{resp}}$, leading to $\mathcal{L}(r^*) = 0$. In Fig. 3**B**, we show an example plot for the loss function. Notably, locally, this plot shows a kink at the global minimum and a no-learning zone for $r < r_b$, similar to the ghost model above. However, in reality, the global loss landscape is far more complicated and no global no-learning zone exists.

As we discuss below, it is also possible to show that a local no-learning zone exists at $r < r_b = \frac{1}{2(T_{\text{delay}}+T_{\text{resp}})}$, as for any smaller $r$ satisfying $r > -r_b$, the system will only output 0 throughout the whole trial, hence a flat loss region exists with the value of 1 for all $r \in [-r_b, r_b]$. Moreover, using geometrical reasoning, we can analytically compute the loss function for $-\frac{1}{2(T_{\text{delay}}+T_{\text{resp}})} \leq r \leq \frac{1}{T_{\text{delay}}+T_{\text{resp}}}$ under the condition that $T_{\text{delay}} \gg T_{\text{resp}}$, and for $r > -\frac{1}{4T}$ and $r < \frac{3}{4T}$ when $T_{\text{delay}} = T_{\text{resp}} = T$. First, we start with the limit that $T_{\text{delay}} = T_{\text{resp}} = T$, and then later consider the long delay limit.

### S2.2.2. ANALYTICAL CALCULATIONS WITH EQUAL DELAY AND RESPONSE TIMES

Noting that $T_{\text{half}} = \frac{1}{2|r|}$ is the half period of the oscillations, we start by considering the regime $-\frac{1}{4T} \leq r \leq \frac{1}{4T}$, in which $T_{\text{half}} \geq 2T$. In this case, $x(t) = 1$ or $0$ for the full $2T$ duration, depending on whether $r$ is positive or negative. In both cases, the loss function is $\mathcal{L}(-\frac{1}{4T} \leq r \leq \frac{1}{4T}) = 1$, *i.e.*, flat.

Next, we consider the interval for which $T \leq T_{\text{half}} \leq 2T$, *i.e.*, $\frac{1}{4T} \leq r \leq \frac{1}{2T}$. Then, for the times $T \leq t \leq T_{\text{half}}$, the model outputs zero, whereas it should have outputted one. For this interval, the loss function then becomes:

$$\mathcal{L}(r) = \frac{T_{\text{half}}}{T} - 1 = \frac{1}{2Tr} - 1 \quad \text{for} \quad \frac{1}{4T} \leq r \leq \frac{1}{2T}. \tag{S18}$$

Finally, we consider the interval $\frac{2T}{3} \leq T_{\text{half}} \leq T$, which corresponds to $\frac{1}{2T} \leq r \leq \frac{3}{4T}$. In this case, the network outputs zero for $T_{\text{half}}$ times (making an error in $T - T_{\text{half}}$ out of these), and one for another $T_{\text{half}}$, only to come back to zero for another $2T - 2T_{\text{half}}$ times. Thus, in total, the network makes mistakes in $3(T - T_{\text{half}})$ out of the $2T$ total times. This leads to the loss function:

$$\mathcal{L}(r) = 3 - \frac{3T_{\text{half}}}{T} = 3 - \frac{3}{2Tr} \quad \text{for} \quad \frac{1}{2T} \leq r \leq \frac{3}{4T}. \tag{S19}$$

Bringing all these together, we arrive at the loss function for $-\frac{1}{4T} \leq r \leq \frac{3}{4T}$ as:

$$\mathcal{L}(r) = \begin{cases} 1 & \text{for} \quad -\frac{1}{4T} \leq r \leq \frac{1}{4T}, \\ \frac{1}{2Tr} - 1 & \text{for} \quad \frac{1}{4T} \leq r \leq \frac{1}{2T}, \\ 3 - \frac{3}{2Tr} & \text{for} \quad \frac{1}{2T} \leq r \leq \frac{3}{4T}. \end{cases} \tag{S20}$$

Here, it is clear that for $r^* = \frac{1}{2T}$, the global minimum is achieved such that $\mathcal{L}(r^*) = 0$. We can also compute the gradient at this point as:

$$\text{At } r = \frac{1}{2T}: \quad \nabla \mathcal{L}(r) = \begin{cases} -2T & \text{from the left,} \\ 6T & \text{from the right.} \end{cases} \tag{S21}$$

Next, given that $r_b = \frac{1}{4T}$, we can compute an effective learning rate using the kink at the global optimum as:

$$\alpha_{\text{LC}}^* = \frac{r^* - r_b}{|\nabla \mathcal{L}(r_+^*)|} = \frac{1}{4T} \frac{1}{6T} = \frac{1}{24} T^{-2}. \tag{S22}$$

Interestingly, creating a limit cycle to solve the delayed response task leads to a better scaling with the trial time. Notably, $\alpha_{\text{ghost}}^*$ in Eq. (S6) has a much higher pre-factor than $\alpha_{\text{LC}}^*$ in Eq. (S22). This means that for small $T$, it may be favorable to learn ghost points, whereas for large $T$, limit cycles can be preferable.

### S2.2.3. ANALYTICAL CALCULATIONS WITH LONG DELAYS

Now, we consider the limit $T_{\text{delay}} \gg T_{\text{resp}}$, and compute the analytical loss function for $-\frac{1}{2(T_{\text{delay}} + T_{\text{resp}})} \leq r \leq \frac{1}{T_{\text{delay}} + T_{\text{resp}}}$.

Firstly, similar to before, the loss function is flat for the region $-\frac{1}{2(T_{\text{delay}} + T_{\text{resp}})} \leq r \leq \frac{1}{2(T_{\text{delay}} + T_{\text{resp}})}$, since $T_{\text{half}} \geq T_{\text{delay}} + T_{\text{resp}}$. In other words, the model outputs either zeros (for $r > 0$) or ones (for $r < 0$) throughout the full window, leading to $\mathcal{L}(r) = 1$.

Next, we consider the case where $T_{\text{delay}} \leq T_{\text{half}} \leq T_{\text{delay}} + T_{\text{resp}}$, *i.e.*, $\frac{1}{2(T_{\text{delay}} + T_{\text{resp}})} \leq r \leq \frac{1}{2T_{\text{delay}}}$. In this case, the network outputs zero for $T_{\text{half}}$, which is longer than $T_{\text{delay}}$ and thus leading to an error contribution for times $T_{\text{half}} - T_{\text{delay}}$. Since $T_{\text{half}} \leq T_{\text{delay}} + T_{\text{resp}}$ and $2T_{\text{half}} \geq 2T_{\text{delay}} \geq T_{\text{delay}} + T_{\text{resp}}$, the network correctly outputs one for the rest of the trial. Hence, the loss function becomes:

$$\mathcal{L}(r) = \frac{T_{\text{half}}}{T_{\text{resp}}} - \frac{T_{\text{delay}}}{T_{\text{resp}}} = \frac{1}{2T_{\text{resp}} r} - \frac{T_{\text{delay}}}{T_{\text{resp}}} \quad \text{for} \quad \frac{1}{2(T_{\text{delay}} + T_{\text{resp}})} \leq r \leq \frac{1}{2T_{\text{delay}}}. \tag{S23}$$

Thirdly, we consider the case $\frac{T_{\text{delay}} + T_{\text{resp}}}{2} \leq T_{\text{half}} \leq T_{\text{delay}}$, *i.e.*, $\frac{1}{2T_{\text{delay}}} \leq r \leq \frac{1}{T_{\text{delay}} + T_{\text{resp}}}$. Since $T_{\text{delay}} \geq T_{\text{half}}$, the model starts outputting ones even before $T_{\text{delay}}$ is reached. Hence, there is a contribution to the loss function for the time duration $T_{\text{delay}} - T_{\text{half}}$. However, since $2T_{\text{half}} \geq T_{\text{delay}} + T_{\text{resp}}$, the model correctly outputs the ones during the response window. Hence, the loss function becomes:

$$\mathcal{L}(r) = 1 - \frac{T_{\text{half}}}{T_{\text{delay}}} = 1 - \frac{1}{2T_{\text{delay}} r} \quad \text{for} \quad \frac{1}{2T_{\text{delay}}} \leq r \leq \frac{1}{T_{\text{delay}} + T_{\text{resp}}}. \tag{S24}$$

Finally, we can consider the case $\frac{T_{\text{delay}}}{2} \leq T_{\text{half}} \leq \frac{T_{\text{delay}}+T_{\text{resp}}}{2}$, *i.e.*, $\frac{1}{T_{\text{delay}}+T_{\text{resp}}} \leq r \leq \frac{1}{T_{\text{delay}}}$. Then, since $2T_{\text{half}} \geq T_{\text{delay}}$, the network will incorrectly output one for the duration $T_{\text{delay}} - T_{\text{half}}$. Moreover, since $2T_{\text{half}} \leq T_{\text{delay}} + T_{\text{resp}}$ and by assumption $T_{\text{resp}} \leq \frac{T_{\text{delay}}}{2}$ (and thereby $T_{\text{resp}} \leq T_{\text{half}}$), the network will output zero incorrectly for the time interval $T_{\text{delay}} + T_{\text{resp}} - 2T_{\text{half}}$. Then, the loss function here becomes:

$$\mathcal{L}(r) = 1 - \frac{T_{\text{half}}}{T_{\text{delay}}} + \frac{T_{\text{delay}} + T_{\text{resp}} - 2T_{\text{half}}}{T_{\text{resp}}} = 2 + \frac{T_{\text{delay}}}{T_{\text{resp}}} - \frac{1}{2rT_{\text{delay}}} - \frac{1}{rT_{\text{resp}}} \quad \text{for} \quad \frac{1}{T_{\text{delay}} + T_{\text{resp}}} \leq r \leq \frac{1}{T_{\text{delay}}}. \tag{S25}$$

Bringing all these together, we arrive at the (local values of the) loss function as:

$$\mathcal{L}(r) = \begin{cases} 1 & \text{for} & -\frac{1}{2(T_{\text{delay}}+T_{\text{resp}})} \leq r \leq \frac{1}{2(T_{\text{delay}}+T_{\text{resp}})}, \\ \frac{1}{2rT_{\text{resp}}} - \frac{T_{\text{delay}}}{T_{\text{resp}}} & \text{for} & \frac{1}{2(T_{\text{delay}}+T_{\text{resp}})} \leq r \leq \frac{1}{2T_{\text{delay}}}, \\ 1 - \frac{1}{2rT_{\text{delay}}} & \text{for} & \frac{1}{2T_{\text{delay}}} \leq r \leq \frac{1}{T_{\text{delay}}+T_{\text{resp}}}, \\ 2 + \frac{T_{\text{delay}}}{T_{\text{resp}}} - \frac{1}{2rT_{\text{delay}}} - \frac{1}{rT_{\text{resp}}} & \text{for} & \frac{1}{T_{\text{delay}}+T_{\text{resp}}} \leq r \leq \frac{1}{T_{\text{delay}}}. \end{cases} \tag{S26}$$

Here, it is clear that for $r^* = \frac{1}{2T_{\text{delay}}}$, the global minimum is achieved such that $\mathcal{L}(r^*) = 0$. We can also compute the gradient at this point as:

$$\text{At } r = \frac{1}{2T_{\text{delay}}} : \quad \nabla\mathcal{L}(r) = \begin{cases} -2\frac{T_{\text{delay}}^2}{T_{\text{resp}}} & \text{from the left,} \\ 2T_{\text{delay}} & \text{from the right.} \end{cases} \tag{S27}$$

Next, given that $r_b = \frac{1}{2(T_{\text{delay}}+T_{\text{resp}})}$, we can compute the distance between the minimum and the no-learning zone as:

$$r^* - r_b = \frac{1}{2T_{\text{delay}}} - \frac{1}{2(T_{\text{delay}} + T_{\text{resp}})} = \frac{T_{\text{resp}}}{2T_{\text{delay}}(T_{\text{delay}} + T_{\text{resp}})} \xrightarrow{T_{\text{resp}} \ll T_{\text{delay}}} \frac{T_{\text{resp}}}{2T_{\text{delay}}^2}. \tag{S28}$$

Then, we can compute the critical learning rate as

$$\alpha_{\text{LC}-\text{as}}^* = \frac{r^* - r_b}{|\nabla\mathcal{L}(r_+^*)|} \approx \frac{T_{\text{resp}}}{2T_{\text{delay}}^2} \frac{1}{2T_{\text{delay}}} = \frac{T_{\text{resp}}}{4} T_{\text{delay}}^{-3}. \tag{S29}$$

In words, if the delay period is significantly longer than the response period, the critical learning rate scales as $\sim O(T_{\text{delay}}^{-3})$. It is worth noting that another candidate for the critical learning rate can be computed from the kink at $r' = \frac{1}{T_{\text{resp}}+T_{\text{delay}}}$, which provides the same scaling.

## S3. Experimental details on empirical evaluations

### S3.1. Simulating RNN models

In this section, we describe how we implemented the update rule for the leaky firing-rate RNN defined in Equation (1). Since the experiments are conducted in discrete time, we use the forward Euler method to update the firing rates:

$$r(t+1) = r(t) + \alpha f(r(t), u(t)) \tag{S30}$$

Here, $r$ denotes the firing rate, $f(\cdot)$ corresponds to the update function defined in Equation (1), and $\alpha$ is the step size, computed as $\alpha = \Delta t / \tau$. The update function can be written more explicitly as:

$$r(t+1) = (1-\alpha)\,r(t) + \alpha\,\tanh(Wr(t) + W^{in}u(t) + b + \epsilon) \tag{S31}$$

With this update rule, we can compute the firing rate at the next time step. The only remaining component is computing the gradients from the predicted output $\hat{o}(t)$ to the target output $o^*(t)$, as described in Sections S3.2 and S3.3.

## S3.2. Task details

Short-term memory is a fundamental cognitive process essential for mammalian survival. Several well-established studies have proposed theories on memory maintenance that have deepened our understanding of short-term memory, as illustrated in Fig. 1. However, while these theories focus solely on memory maintenance and response processes, they do not fully explain the underlying mechanisms. This knowledge gap raises a critical question: Which mechanisms govern short-term memory in the brain? To address this question, we investigate short-term memory through two key properties: memory maintenance and information processing. Accordingly, we conduct experiments on two well-established neuroscience tasks: the **delayed activation** and **delayed cue-discrimination** tasks. We determined these tasks because the first task exclusively examines memory maintenance, while the second incorporates both information processing and the retention of input stimuli to generate the relevant response. Furthermore, to enhance transparency and ensure the reproducibility of our experiments, we provide a detailed outline of all task properties in this section.

### S3.2.1. DELAYED ACTIVATION TASK

The delayed activation task allows us to isolate the memory component by assuming that cue processing has already occurred. In this setup, the task consists of three distinct periods: the *delay period* ($T_{\text{delay}}$), the *reaction period* ($T_{\text{resp}}$), and the optional *post-reaction period* ($T_{\text{post}}$). Since the cue period is omitted, the input remains zero throughout the trial, making the task entirely dependent on the intrinsic dynamics of the recurrent model rather than external stimuli.

The expected ground truth responses, denoted as $O^* = \{o_1^*, o_2^*, o_3^*, ..., o_T^*\}$, are defined over the total duration $T = T_{\text{delay}} + T_{\text{resp}} + T_{\text{post}}$ as follows:

$$o_t^* = \begin{cases} 1, & \text{if } t \in T_{\text{resp}} \\ 0, & \text{otherwise} \end{cases} \tag{S32}$$

This task is inherently simpler than the delayed cue-discrimination task, as it isolates the challenge of maintaining information in memory without requiring cue processing. By setting the input $u$ to zero at all times, the model must rely solely on its internal state dynamics to correctly generate the expected response during the reaction period. The optional post-reaction period allows for additional investigation into how the network dynamics stabilize after completing the required response.

### S3.2.2. DELAYED CUE-DISCRIMINATION TASK

In contrast to the delayed activation task, the delayed cue-discrimination task involves two input channels and two output channels. It consists of four distinct periods: the *cue period* ($T_{\text{in}}$), *delay period* ($T_{\text{delay}}$), *reaction period* ($T_{\text{resp}}$), and an optional *post-reaction period* ($T_{\text{post}}$). This task allows us to investigate both stimulus processing and memory components simultaneously.

The input cue for the delayed cue-discrimination task, denoted as

$$U = \{(u_1^1, u_1^2), (u_2^1, u_2^2), (u_3^1, u_3^2), ..., (u_T^1, u_T^2)\}$$

is defined over the total duration $T = T_{\text{in}} + T_{\text{delay}} + T_{\text{resp}} + T_{\text{post}}$ as follows:

$$u_t = \begin{cases} (0, 1) \text{ or } (1, 0), & \text{if } t \in T_{\text{in}} \\ (0, 0), & \text{otherwise} \end{cases} \tag{S33}$$

The expected ground truth outputs, denoted as

$$O^* = \{(o_1^{*1}, o_1^{*2}), (o_2^{*1}, o_2^{*2}), (o_3^{*1}, o_3^{*2}), ..., (o_T^{*1}, o_T^{*2})\}$$

are defined as:

$$o_t^* = \begin{cases} u_{T_{in}}, & \text{if } t \in T_{\text{resp}} \\ (0,0), & \text{otherwise} \end{cases} \tag{S34}$$

This task requires the model to process the cue presented during the cue period, maintain it through the delay period, and then generate the appropriate response during the reaction period. The optional post-reaction period allows for additional investigation into how the network dynamics stabilize after completing the required response.

### S3.3. Training details

To ensure the reproducibility of our experiments, we provide a comprehensive description of the training configurations used in this study. All recurrent neural network (RNN) architectures were implemented using the PyTorch framework. Unless stated otherwise, models were trained using the stochastic gradient descent (SGD) with the momentum value of $0.9$ and weight decay value of $10^{-7}$. Most of the experiments were conducted on computing systems equipped with an Intel i9-10900X CPU and an Apple M2 CPU. To further enhance transparency and support future research, we will publicly release our complete training pipeline on GitHub.

Training with long delay intervals, i.e., $T_{\text{delay}} \gg T_{\text{resp}}$, presents challenges when compared to shorter delay times due to the model's tendency to learn to always output 0. This issue arises from an inherent imbalance in the loss function: since the model is expected to output 0 for the majority of the trial and only 1 during the response period, a naive mean-squared error (MSE) loss would favor minimizing errors in the dominant interval, leading to trivial solutions where the model outputs 0 at all times. This imbalance is aggravated when the total trial duration $T_{\text{in}} + T_{\text{delay}} + T_{\text{post}}$ is significantly greater than $T_{\text{resp}}$.

To mitigate this issue, we trained all RNNs using a weighted MSE loss, where different time intervals were assigned corresponding weights. The weight function was defined as follows:

$$w_{MSE} = \begin{cases} 1 - \frac{10}{T_{\text{total}}}, & \text{if } t \in T_{\text{resp}} \\ \frac{10}{T_{\text{total}}}, & \text{otherwise} \end{cases} \tag{S35}$$

This weighting scheme ensures that the loss function does not disproportionately favor the dominant 0 responses and appropriately emphasizes the importance of correct responses during the reaction period.

### S3.4. Details on large-scale experiments

In this study, we conducted several short-term memory experiments to investigate underlying memory mechanisms. Specifically, we examined rank-2 RNNs trained on the delayed activation task, both with and without a post-reaction period (Fig. 2, 3, and S1). To extend our observations to full-rank RNNs, we increased task complexity by training models on the delayed cue-discrimination task, again with and without a post-reaction period (Fig. 4, S3, and S4). Additionally, to explore the role of the time constant $\tau$ and memory-related components, we evaluated full-rank RNNs under various configurations (Fig. S5 and S6). Further implementation and training details are provided in Sections S4 and S3.3. As a summary, the configurations and number of total runs for each experiment are presented below:

*Table S1.* Large-scale experiment configurations and the number of total runs.

| RNN | Task | Learning Rate | $T_{\text{delay}}$ | # Seeds | # Experiments |
|---|---|---|---|---|---|
| Rank-2 | Delayed act. w/wo $T_{\text{post}}$ | $5 \times 10^{-3}$ | $30\,\text{ms}$ | 100 | 1600 |
| Full | Cue-disc. wo/ $T_{\text{delay}}$ | $\log[10^{-3}, 1]$ (20) | $T_{\text{resp}} = \text{lin}[20, 400]\,\text{ms}$ (20) | 20 | 8000 |
| Full | Cue-disc. w/ longer $\tau$ | $\log[10^{-3}, 10^{-1}]$ (10) | $\text{lin}[0, 800]\,\text{ms}$ (21) | 20 | 4200 |
| Full | Cue-disc. wo/ $T_{\text{post}}$ | $\log[10^{-3}, 10^{-1}]$ (15) | $\text{lin}[0, 400]\,\text{ms}$ (21) | 100 | 31500 |
| Full | Cue-disc. wo/ $T_{\text{post}}$ | $\log[10^{-1.6}, 10^{-0.5}]$ (15) | $\text{lin}[0, 400]\,\text{ms}$ (21) | 100 | 31500 |
| Full | Cue-disc. w/ $T_{\text{post}}$ | $\log[10^{-3}, 10^{-1}]$ (15) | $\text{lin}[0, 400]\,\text{ms}$ (21) | 20 | 6300 |

S3.4.1. DETECTING LEARNED MECHANISMS SUBSERVING SHORT-TERM MEMORY IN RNNS

We designed two metrics—Reaction Accuracy (RA) and Reaction Reliability (RR)—to evaluate the performance of our recurrent neural network (RNN) models. Reaction Accuracy quantifies the model's ability to discriminate the input cue by identifying the class corresponding to the maximum output channel at each time step, while Reaction Reliability assesses the degree of signal distinction between the classes.

More formally, let $O \in \mathbb{R}^{N \times 2}$ represent the model's output vector over the reaction interval, and $G \in \mathbb{R}^{N \times 1}$ denote the ground truth labels over the same interval, where $N$ is the total number of time steps.

**Reaction Accuracy (RA):**  This metric assesses how accurately the model produces the correct response during the response period $(T_{\text{resp}})$. It is computed as follows:

$$RA = 1 - \frac{1}{N} \sum_{t=1}^{N} \left| \arg \max_{c}(O_t^c) - G_t \right| \tag{S36}$$

Here, $\arg \max_c(O_t^c)$ returns the index of the output channel with the highest activation at time step $t$, and $G_t$ denotes the ground truth class label at that time.

**Reaction Reliability (RR):**  This metric complements Reaction Accuracy by not only evaluating whether the model selects the correct response during the response period $T_{\text{resp}}$, but also by measuring how confidently it does so. It is computed as follows:

$$RR = \frac{1}{N} \sum_{t=1}^{N} \frac{O_{t,G_t}}{\sum_{c=1}^{2} O_t^c} \tag{S37}$$

Here, $O_{t,G_t}$ denotes the model's output corresponding to the ground truth class $G_t$ at time step $t$, and the denominator normalizes this value by the total output across both output channels at that time step.

**Mechanism Discrimination Index (MDI):**  The two metrics, RA and RR, allow us to determine whether the model has successfully learned the task. If a model fails to learn, we categorize its behavior as either 'Unstable Training' or 'Final Model Failure', as illustrated in Fig. S3**A-B** and Fig. S4**A**. However, these metrics alone do not clearly distinguish between limit cycles and slow-point manifolds (see Fig. S3**A-B** and Fig. S4**C-D**). Therefore, to reliably differentiate these two mechanisms, we further analyze the frequency distributions of correlation coefficient matrices computed from the first two principal components of the models' firing rates, as shown in Fig. S3**C**.

Since limit cycles closely approximate slow-point manifolds during the trial period, it is challenging to distinguish them based solely on firing rates observed within that interval. To overcome this limitation, we extend the analysis window by concatenating the post-reaction period during inference, thereby revealing the underlying dynamical structure of the firing rates, as illustrated in Fig. S4**C-D**. Next, we apply principal component analysis (PCA) to the firing rates and extract their first two principal components (Fig. S4). We then compute the Pearson product-moment correlation coefficient matrix from these principal components.

Due to the repetitive structure of limit cycles, their correlation coefficient matrices exhibit predominantly high-frequency components. In contrast, slow-point manifolds produce correlation matrices characterized by consistently high correlations, corresponding predominantly to low-frequency components. After calculating the frequency components, we define a hyperparameter to establish a threshold for discrimination between low-frequency and high-frequency, and compute the Mechanism Discrimination Index (MDI) as follows:

$$MDI = \frac{PSD_{lf}}{PSD_{total}} \tag{S38}$$

where $PSD_{lf}$ is the power spectral density of low-frequency components, and $PSD_{total}$ is the total power spectral density. Using this approach, we reliably discriminate between limit cycles and slow-point manifolds, as demonstrated in Fig. S3**C**.

As a summary, we defined four categories to classify the outcomes of RNN training:

- **Final Model Failure:** An RNN is considered to have failed to converge if its mean accuracy and reliability have never previously reached or exceeded 0.8 (Fig. S4**A**).

- **Unstable Training:** Training is classified as unstable if, after initially achieving accuracy and reliability values above 0.8, the performance subsequently falls below 0.6 for more than 10% of the epochs following the initial success (Fig. S4**B**).

- **Limit Cycle:** An RNN is classified as exhibiting a limit cycle if its mean accuracy and reliability during the final 10% of epochs are $\geq 0.8$, and the Mechanism Discrimination Index (MDI) is less than 0.5 (Fig. S4**C**).

- **Ghost Point:** An RNN is classified as exhibiting a ghost point if its mean accuracy and reliability during the final 10% of epochs are $\geq 0.8$, and the Mechanism Discrimination Index (MDI) is greater than or equal to 0.5 (Fig. S4**D**).

The first two categories collectively constitute the "Not Learned" classification.

S3.4.2. PLOTTING THE SCALING LAWS

To investigate the scaling relationship between the learnable delay interval and the learning rate, we performed the following analysis.

First, for each learning rate, we extracted the model's performance across different delay intervals from the large-scale experiments (Fig. 4**B, E**). We then fit a saturating function—either an exponential saturation or a sigmoid—to the performance curves over delay (Fig. S3**D, E**). This yielded a cutoff delay value $a^*$, representing the point at which performance saturates or exhibits a sharp transition. Specifically, the exponential saturation fit was defined as:

$$f(x; x_0, \alpha) = \begin{cases} 0 & \text{if } x < x_0 \\ 1 - \exp(-\alpha(x - x_0)) & \text{otherwise} \end{cases} \tag{S39}$$

The fits were applied to performance metrics (e.g., reaction accuracy or reliability) over delay intervals $x$, using `scipy.optimize.curve_fit`.

Next, we analyzed how the cutoff delay values $a^*$ scale with learning rate $\alpha$. We performed a log-log linear regression of the form:

$$\log(\alpha) = \beta \log(a^*) + \log(A) \tag{S40}$$

The critical exponent $\beta$ was estimated using `scipy.stats.linregress` and validated by an additional log-log fit via `curve_fit`. To estimate confidence intervals, we performed Monte Carlo sampling using the parameter covariance matrix. The resulting fit and its 95% confidence interval were visualized on log-log axes, as shown in Fig. 4**C, F** and Fig. S3**F**.

This analysis enabled us to extract empirical scaling laws for slow-point manifolds and limit-cycles, which were then compared to theoretical predictions.

S3.4.3. PLOTTING FIRING RATES

To visualize firing rates, we collect the hidden states of the leaky firing-rate RNN during inference. To reveal the underlying dynamical structure, we extend the trial by appending a post-reaction interval ($T_{\text{post}}$) that is nine times longer than the original duration, defined as $T = T_{\text{in}} + T_{\text{delay}} + T_{\text{resp}}$.

After collecting the full firing rate trajectory, we sort neurons based on the timing of the peak activation in their absolute firing rates. Specifically, each neuron's activity is scanned to identify the time step at which its firing rate reaches its maximum value. Neurons that do not exceed a predefined activation threshold are pushed to the end of the ordering. This procedure ensures that neurons are visualized in the order in which they become active, highlighting sequential structure in the network's dynamics.

# S4. Reproduction details

**Figure 2:** To investigate the emergent mechanisms—specifically, slow-point manifolds and limit cycles—we conducted experiments using low-rank recurrent neural networks (RNNs) (**Background**). Given that a minimum of two dimensions is required for the formation of limit cycles, we set the low-rank dimensionality to $K = 2$. The delayed activation task was chosen as the training paradigm. The learning rate was initialized at $\alpha = 10^{-2}$. Additionally, we set the time decay constant to $\tau = 10$ ms, the time step to $\Delta t = 5$ ms, the delay interval to $T_{\text{delay}} = 150$ ms, and the response interval to $T_{\text{resp}} = 50$ ms. The architecture was trained for $150,000$ epochs with a total of $N = 100$ neurons. All weights and biases, as described in Eq. (2), were optimized throughout training. For this experiment, noise was set to $\epsilon = 0$, and firing rates were initialized from a multivariate Gaussian distribution with mean $\mu_r = 0$ and standard deviation $\sigma_r = 0.1$. Training was performed using stochastic gradient descent (SGD) with PyTorch's default hyperparameters.

**Figure 3:** This figure highlights the impact of even slight changes in task configuration on network dynamics. To demonstrate this, we trained identical RNNs with $N = 100$ neurons and a rank of $K = 2$, by increasing the post-reaction period interval from $T_{\text{post}} = 0$ ms to $T_{\text{post}} = 30$ ms. We applied a similar configuration to Fig. 2**A**, except for setting $T_{\text{delay}} = 30$ ms, $T_{\text{resp}} = 10$ ms, and $\alpha = 10^{-3}$ for the experiments in Fig. S1**A-B**, training the networks for $10^6$ epochs. For the experiments in Fig. S1**C**, we set $\alpha = 5 \times 10^{-3}$ and trained the networks for $500,000$ epochs while varying the delay interval as $T_{\text{delay}} \in \{30, 40, 50, 60, 70\}$ ms. Moreover, SGD is used with the default hyperparameters of PyTorch.

**Figure 4:** In large-scale RNN experiments, we developed a primary training pipeline to systematically investigate the effects of learning rate, delay interval length, and post-reaction interval. Apart from these variations, all other training configurations in Fig. 4 were set as follows: number of neurons $N = 100$, noise initialization drawn from a Gaussian distribution with mean $\mu_\epsilon = 0$ and standard deviation $\sigma_\epsilon = 10^{-3}$, and firing rate initialization drawn from a Gaussian distribution with mean $\mu_r = 0$ and standard deviation $\sigma_r = 10^{-1}$. Additionally, task-related parameters were fixed, with the cue interval set to $T_{\text{in}} = 30$ ms, the reaction interval to $T_{\text{resp}} = 50$ ms, the time decay constant to $\tau = 10$ ms, and the time step to $\Delta t = 5$ ms. Since lower learning rates require longer training durations, we implemented an adaptive schedule for the number of training epochs based on the learning rate. The number of epochs was determined using the following equation:

$$\text{num\_epochs} = \max(10^3, \frac{30}{\alpha}) \tag{S41}$$

The delay interval was gradually increased in increments of 20 ms, covering the range $T_{\text{delay}} \in [0, 420]$ ms. In our initial experiments, learning rates were sampled from a logarithmic scale with equal spacing in the range $\alpha \in [10^{-3}, 10^{-1}]$. To further investigate the effects of intermediate learning rates, we additionally sampled more values logarithmically within the range $\alpha \in [10^{-1.6}, 10^{-0.5}]$ (see Table S1 for details). Besides, Fig. 4**C,F** presents the estimated cutoff delay values, which are described in detail in Section **2.5**. Moreover, in Fig. **H, I**, we constructed a phase diagram of emergent mechanistic phases derived from the results shown in Fig. 4**C-D**.

**Figure S2:** We analyze the emergence of the limit cycle model in the delayed activation task through analytical methods (**Supplementary 1.2**). Figure S2**A** illustrates the vector field of the dynamical system, which is derived from Eq. (S15) with $r = 0.1$. Additionally, Figure 4**G** presents two loss curves: the analytical curve is obtained from Eq. (S17), while the numerical curve is computed using Eq. (S20), with parameters set to $T_{\text{delay}} = 60$ ms, $T_{\text{resp}} = 10$ ms, and $dt = 10^{-2}$. Finally, Figure S2**B** is constructed by plotting analytically derived scaling laws, with the response interval set to $T_{\text{resp}} = 10$ ms.

**Figure S3:** This figure builds on the results presented in Fig. 4 to illustrate the RA and MDI metrics. Additionally, it highlights the scaling laws associated with slow-point manifolds observed in the large-scale RNN experiments.

**Figure S4:** We randomly sampled the example classes of dynamical systems from the large-scale RNN experiments.

**Figure S5:** In this experiment, we use the same training configuration as in Fig. 4, except for the delay intervals and the value of $\tau$. We set $\tau = 20$ ms and sampled $T_{\text{delay}}$ from the range $[0, 840]$ ms in increments of 40 ms. (See Table S1 for more details.)

**Figure S6:** In this figure, the only differences from the large-scale experiments are that the number of training epochs is set to 3000, $T_{\mathrm{delay}} = 0$ ms, and the response interval $T_{\mathrm{resp}}$ is varied instead. This setup allows us to examine the importance of the memory component. (See Table S1 for more details.)

The code to reproduce these figures is publicly available at https://github.com/fatihdinc/dynamical-phases-stm, the data jointly with the code at https://doi.org/10.5281/zenodo.15529757.

# Figures and captions

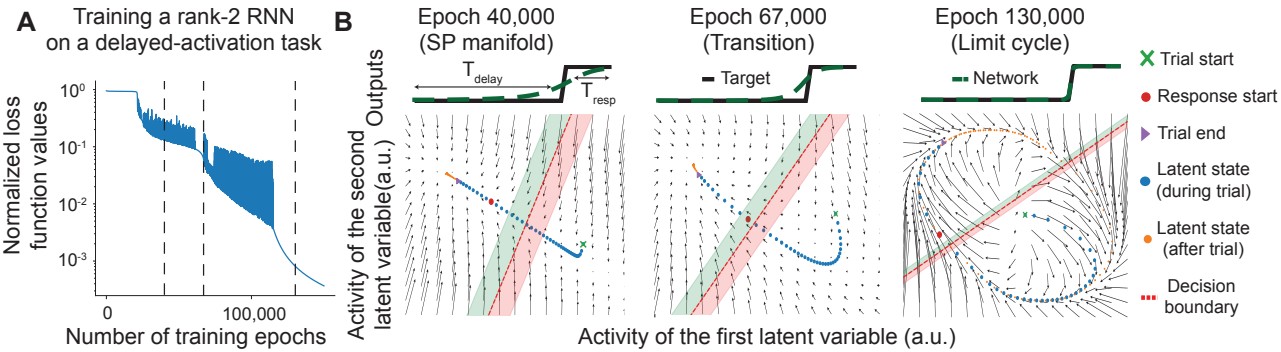

*Figure S1.* **RNNs can switch between distinct short-term memory mechanisms during training. A** To illustrate that different mechanisms can be employed by dynamical systems to delay their responses, we trained a rank-2 RNN to perform the delayed activation task. In this task, networks initialized at a given state (**Methods**) should delay their output by $T_{\text{delay}}$. Parameters: $N = 100$ neurons, $\tau = 10$ms, $\Delta t = 5$ms, $T_{\text{delay}} = 150$ms, $T_{\text{resp}} = 50$ms, $\alpha = 10^{-2}$ for stochastic gradient descent using otherwise the default parameters in `PyTorch` (Paszke et al., 2017). The dotted lines correspond to epochs, for which the latent dynamical systems are illustrated in Panel (**B**). **B** During the training, two distinct mechanisms emerge in the RNN, which we probe by plotting the two-dimensional latent dynamical systems. *Left.* Early in the training, the network lowers the loss values by designing a line of slow-points (referred to as SP manifold), towards which other stats are attracted. The latent states after the trial concludes end up in a final attractive fixed-point. *Middle.* As training progresses, the SP manifold slowly dissolves into a circular pattern, which we deem as a transient mechanism before the network settles into the final solution. *Right.* After further training, a limit cycle emerges in the latent dynamical system. In this structure, even after the trial concludes, latent states continue to evolve in a periodic manner. The shaded areas around the decision boundary correspond to network outputs within $0.75$ (outer green boundary) and $0.25$ (outer red boundary).

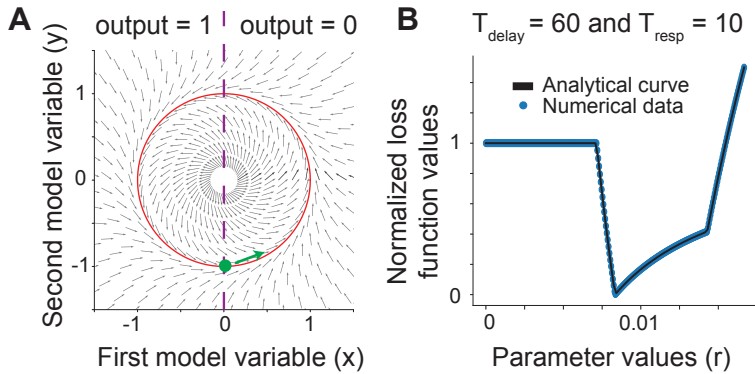

*Figure S2.* **A theoretical scaling analysis of short-term memory mechanisms via toy models. A** The illustration of the flow map corresponding to the toy limit cycle model for $r = 0.1$. The red circle denotes the attractive limit cycle, the purple line corresponds to the decision boundary of the output, and the green circle denotes the starting point of the state. **B** An example loss curve for the limit cycle toy model, with analytical curve and empirical data points. Note the kink at the global minimum. **C** The different dependencies of the effective learning rate for two types of mechanisms create a phase space as a function of delay and learning rate values. In general, there are three main phases of short-term memory mechanisms for large $T_{\mathrm{delay}} \gg T_{\mathrm{resp}}$. For this figure, we picked $T_{\mathrm{resp}} = 10$ms.

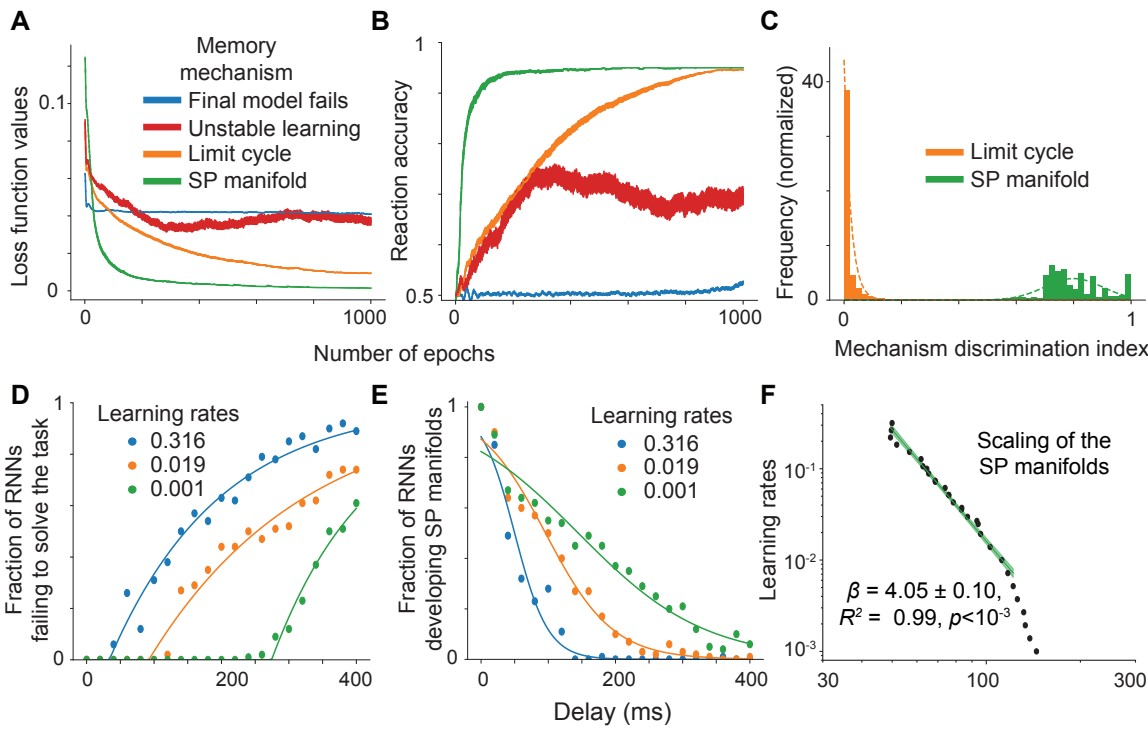

*Figure S3.* **Details of the large-scale RNN experiment. A** Different memory mechanisms exhibit distinct loss curves during training. While limit cycles and slow-point manifolds successfully converge to local minima, the "Final Model Failure" and "Unstable Learning" classes do not achieve stable convergence due to high learning rates. **B** As further illustrated, only the limit cycle and slow-point manifold mechanisms achieve consistently high reaction accuracies. **C** The Mechanism Discrimination Index (MDI) (see **Methods**) allows reliable differentiation between limit cycles and slow-point manifolds, even though their dynamics appear similar during the trial period. **D** Lowering the learning rate enhances the model's ability to learn tasks with longer delay intervals. The solid curve represents the corresponding exponential saturation fit (see **Methods**). **E** Additionally, decreasing the learning rate increases the likelihood of models adopting slow-point manifolds. **F** As predicted by analytical models, results from the large-scale RNN experiments exhibit a similar power-law relationship ($T_{\text{delay}}^{-4} \geq T_{\text{delay}}^{\beta} \geq T_{\text{delay}}^{-5}$) for learning slow-point manifolds for large learning rates. However, for larger delays ($T_{\text{delay}} > 100ms$), these solutions preferentially suppressed, presumably in favor of the limit cycle based solutions.

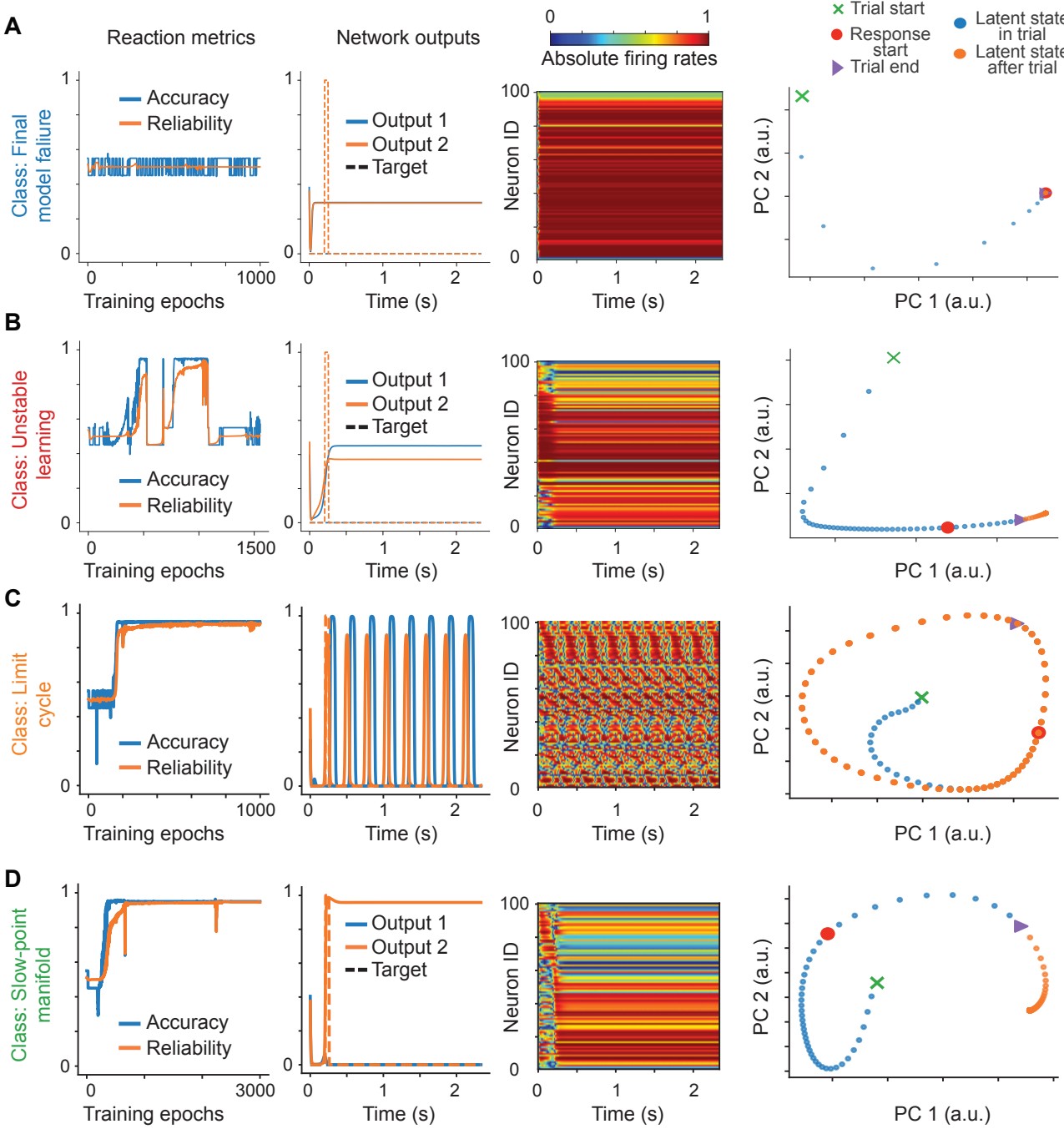

*Figure S4.* **The four main classes identified in the large-scale RNN experiment.** This figure illustrates representative examples from each of the four primary classes: **A** Final model failure, **B** Unstable learning, **C** Limit cycle, and **D** Slow-point manifold. We focused on trained models from experiments with a delay interval of 180 ms, which was specifically selected because it exhibits all four classes, as illustrated in Fig. 4**B**. The first column presents the reaction metrics, *i.e.*, Reaction Accuracy and Reaction Reliability (see **Methods**). The second column shows the network's outputs alongside the corresponding ground truth signals. The third column displays the firing rates of individual neurons. The final column illustrates the first two principal components derived from these firing rates. Training parameters used for each representative example are as follows: **A** $\alpha = 0.1$, seed = 93; **B** $\alpha \approx 0.01931$, seed = 41; **C** $\alpha = 0.1$, seed = 82; and **D** $\alpha = 0.01$, seed = 85. Additional parameters shared across all models are $T_{\text{in}} = 30$ ms, $T_{\text{delay}} = 180$ ms, $T_{resp} = 50$ ms, $\Delta t = 5$ ms, $\tau = 10$ ms, $\sigma_{\text{noise}} = 0.001$, $\sigma_{\text{r}} = 0.1$, and batch size per channel = 64 (see **Methods**).

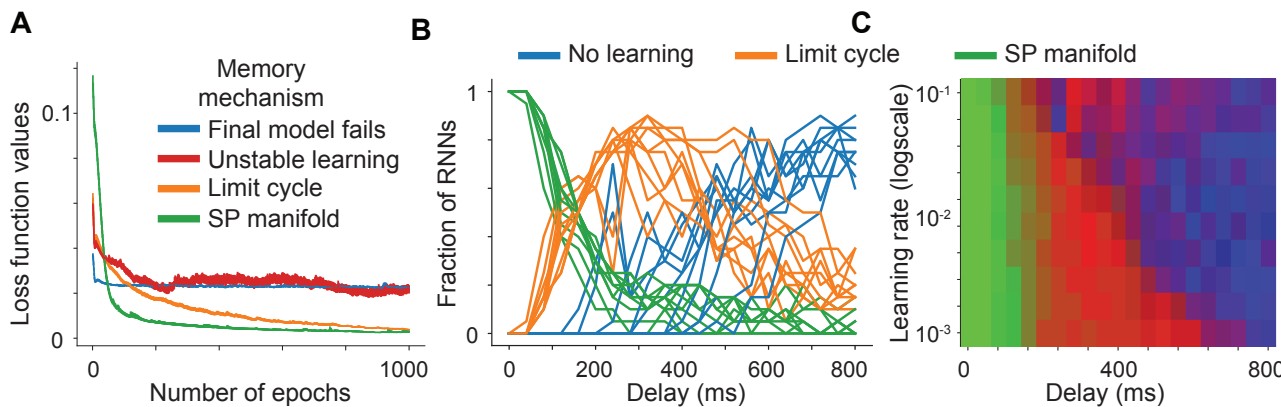

*Figure S5.* **RNN experiments with increased** $\tau$**.** In this figure, we examine the effect of increasing the time constant to $\tau = 20$ ms while keeping all other hyperparameters identical to those used in the large-scale delayed cue-discrimination task experiments (Fig. 4). These experiments reveal that increasing $\tau$ does not alter the observed scaling laws. **A** As in Fig. S3, we observe similar loss curve patterns across all classes under the larger $\tau$ setting. **B** Consistent with previous results, the distribution of dynamical classes across different random seeds remains largely unchanged with the increased $\tau$. **C** The phase diagram for $\tau = 20$ ms shows scaling laws consistent with those observed in the smaller $\tau$ regime.

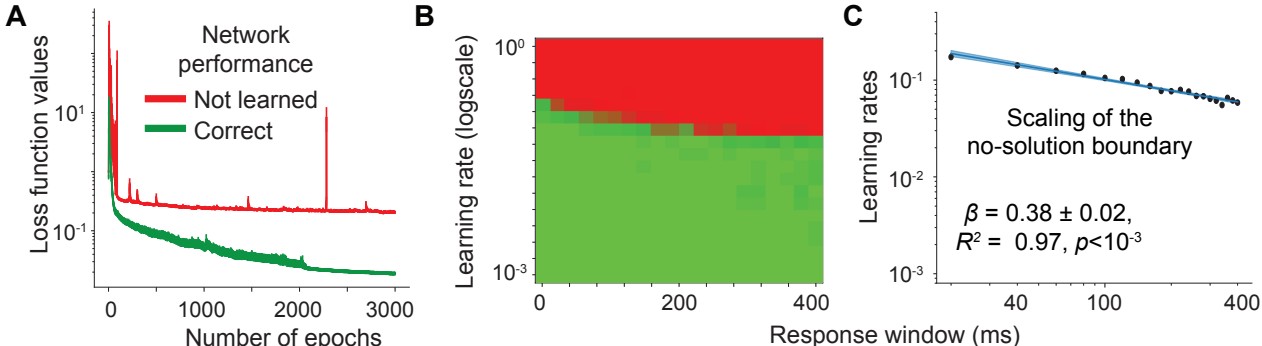

*Figure S6.* **Scaling laws disappear in fixed-response task experiments with RNNs.** In this experiment, we remove the delay period from the delayed cue-discrimination task to isolate the effect of memory and assess its role in scaling laws. As shown in the figure, the removal of the delay component eliminates the previously observed scaling behavior, highlighting memory as a critical factor underlying these laws. **A** As in previous tasks, we observe qualitatively similar loss curves between *not learned* and *correct* networks. **B** Phase diagram showing the classification of *not learned* and *correct* networks in the fixed-response task. The x-axis represents the length of the response interval, during which the model is required to generate a consistent output over time. Notably, removing the delay period (and thus the memory requirement) leads to a substantial loss of scaling structure. **C** We compute the scaling law using the same methodology as in Fig. 4**C,F** and Fig. S3**F**.

