# OpenReview forum: "Dynamical phases of short-term memory mechanisms in RNNs"
_ICML.cc/2025/Conference — ICML 2025 poster_

### Official Review · Reviewer_59aj · 2025-03-12

**Overall Recommendation:** 3

**Summary:**

The paper investigates the strategies that recurrent neural networks (RNNs) use to maintain short-term memories via sequential firing. The authors trained low-rank and full-rank RNNs on delay-response tasks and identified two distinct mechanisms: slow-point (SP) manifolds and limit cycles. They found that introducing a post-response period significantly biases the strategy toward limit cycles. Additionally, they derive a scaling law relating the critical learning rate to the delay period length.

**Claims And Evidence:**

Their major claim was supported mostly by their results on artificial RNNs, but the biological relevance of these mechanisms remains unclear, especially the usage of limit cycles.

**Essential References Not Discussed:**

Not to my knowledge.

**Experimental Designs Or Analyses:**

Yes. The experiments include both low-rank and full-rank RNNs, as well as large-scale training with over 35,000 networks, ensuring robust findings within artificial RNNs.

**Methods And Evaluation Criteria:**

yes

**Other Comments Or Suggestions:**

No

**Other Strengths And Weaknesses:**

- Strengths:
      - Rigorous mathematical derivation.
      - Large-scale empirical validation with 35,000+ trained RNNs.
      - Acute to realize that traditional RNNs can be different from brain doing task by post-action period.
- Weaknesses:
      - Weak connection to biological plausibility and experimental neuroscience.
      - Limited discussion of hyperparameter sensitivity and generalizability.

**Questions For Authors:**

1. What do you think would be the biological correspondence of the learning rate in your model?
2. In Figure 5, the network needs to maintain feature-specific information rather than a featureless memory. How do SP manifolds or limit cycles encode and transmit feature representations?

**Relation To Broader Scientific Literature:**

The paper is well-situated within the computational neuroscience and machine learning literature, particularly in the study of RNN dynamics. The discussion of memory maintenance mechanisms in artificial networks is insightful. However, the connection to empirical neuroscience is underdeveloped.

**Theoretical Claims:**

I did not verify every detail of their mathematical arguments, I reviewed the toy model derivations, and they appear to be correct.

---

> ### Author Rebuttal · Authors · 2025-03-28
>
> We thank the reviewer for their constructive feedback, as well as their recognition of our methodological rigor and insightful contributions to understanding short-term memory mechanisms in artificial networks. Please find our responses below:
>
> **Q1** While learning rate is an abstract optimization hyperparameter, prior literature indicates that it can be influenced by both external and internal biological factors. See below for a brief discussion that we will add:
>
> > Interestingly, our findings that longer delays require smaller learning rates parallels observations in neuroscience, where tasks with longer temporal gaps between cues and outcomes often pose greater credit assignment challenges. During cue-reward delays, dopamine activity has been suggested as a mechanism for solving credit assignment through eligibility traces. For instance, synaptic-level eligibility traces have been evidenced by dopamine's role in potentiating synaptic changes within precise temporal windows (Shindou et al., 2019; Yagishita et al., 2014). Additionally, dopamine may facilitate credit assignment through activity ramps that increase as subjects approach rewards (Fiorillo et al., 2003, Mikkael et al., 2022, Krausz et al., 2023). But more explicitly, a recent study (Coddington et al., 2023) showed that dopamine can directly modulate learning rates to support effective decision-making. Similarly, our results align with studies showing that the duration of the inter-trial interval—analogous to our post-reaction time—can also affect learning rates. Specifically, another neuromodulator, serotonin, has been found to modulate learning rates following long inter-trial intervals (Iigaya et al., 2018).
>
> Thank you for this exciting connection that significantly improves the impact of our work for the neuroscience community.
>
> **Q2** We appreciate this insightful question. Our task was intentionally designed to isolate memory maintenance mechanisms, independent of representational content, to expose the underlying structure of the solution space (e.g., phase transitions). In tasks like delayed cue discrimination (Fig. 5), inputs determine the initial condition of the latent state, which then evolves along either a slow manifold or a limit cycle. In both scenarios, the identity of the cue is encoded in the trajectory, e.g., distinct slow-point manifolds.
>
> How feature representations are integrated into these mechanisms is a compelling direction for future work. For instance, one could introduce a contextual variable that modulates the delay period for the same stimulus. This would allow investigation into whether the RNN reuses its existing slow point or limit cycle structures, or develops new attractors for each context. We view questions like these as natural next steps—and our publicly available dataset of trained RNNs is well-suited for such follow-up studies. We will add a paragraph to the outlook about this interesting future direction.
>
> **Weakness: hyperparameter sensitivity and generalizability** In the revised draft, we include a new experiment in which we double the neuronal time constant $\tau$ (one of the hyperparameters in the RNN architecture). The results remain virtually unchanged, with the same phase diagram emerging. We will report these findings in a supplementary figure. We also note that a recent work (Park et al. 2025; ICLR 2025) has empirically found a similar phase diagram of algorithmic strategies, though in a completely different context.
>
> While testing all hyperparameters is computationally infeasible given the scale of our experiments, we now include the following paragraph in the Discussion to explicitly state this limitation and outline future directions:
>
> > In this work, primarily constrained by the immense compute required (about a month of a standard GPU time), we fixed architectural parameters and activation functions, which is a limitation of this work. Future work could explore the effects of different activation functions, network size (although recent results suggest that larger networks may reduce efficient learning rates; see Dinc et al., 2025), self attention-like mechanisms, and short-term synaptic plasticity on the emergence of different memory strategies. We conjecture that, since the toy models highlight the fundamental constraints associated with the latent dynamical systems learned within these architectures, phase diagrams we observe in this work may present universally across models (see also Park et al., 2025).
>
> **Generalizability and relevance of the toy models** We agree that this is central to the broader impact of our work and should have been stated more clearly. Please also see our response to Reviewer CYoj.
>
> **Final remarks** Please let us know if any further clarifications are needed. We sincerely appreciate the reviewer’s feedback and support. We agree that stronger alignment with neuroscience is important and have made necessary revisions.

---

### Official Review · Reviewer_CYoj · 2025-03-14

**Overall Recommendation:** 2

**Summary:**

This paper analyzes the emergent mechanisms of short-term memory maintenance in task-optimized recurrent neural networks. The paper presents an analysis of a toy model and performs large-scale experiments to show that similar features emerge in actual task-optimized networks.

**Claims And Evidence:**

The theory and numerical experiments are each separately well-executed and interesting, but their connection is difficult to pin down. Typically, theoretical analysis of RNNs is designed to be directly comparable with the outcomes of numerical experiments. However, in this work, the theory seems to target a different model entirely. While the experiments exhibit some qualitative similarities to the theoretical toy model, they feel somewhat disjointed, making it challenging to establish a clear link between theory and practice.

**Essential References Not Discussed:**

None.

**Experimental Designs Or Analyses:**

The RNN experiments make sense, although I did not look at the code in detail.

**Methods And Evaluation Criteria:**

Yes, somewhat (see above).

**Other Comments Or Suggestions:**

For completeness, the introduction should acknowledge that many neuroscientists propose a synaptic basis for working memory, which relies on mechanisms distinct from those presented in the authors' work. For example:

- https://www.cell.com/trends/trends/cognitive-sciences/fulltext/S1364-6613(15)00102-3
- https://journals.plos.org/ploscompbiol/article?id=10.1371/journal.pcbi.1010776
- https://www.nature.com/articles/s44271-023-00027-8?fromPaywallRec=false

**Other Strengths And Weaknesses:**

The work is not particularly novel, as studying attractors as short-term memory mechanisms in RNNs is one of the oldest problems in computational neuroscience. I appreciate the attempt at theory, although as I explain above the connection between the numerical experiments and the theory developed is unclear to me.

**Questions For Authors:**

N/A

**Relation To Broader Scientific Literature:**

This paper would be of interest to computational neuroscientists, as well as deep learning researchers working on interpretability.

**Theoretical Claims:**

Yes, all of them.

---

> ### Author Rebuttal · Authors · 2025-03-28
>
> We thank the reviewer for their thoughtful comments and for recognizing that both the theoretical and empirical components are well-executed. We understand the main concerns to be (1) the clarity of the connection between theory and experiments, and (2) the perceived novelty of our contributions. We have revised the manuscript accordingly and address both below.
>
> **(1)** Due to space constraints, we had to condense this connection in the original submission. Here, we clarify the rationale and why toy models are *directly* relevant for the RNNs.
>
> **In short**, consider the models in Eq. (S1) and (S15) of the main text. These approximate the latent trajectory of a rank-one and rank-two RNN around a local minima, respectively. The scaling laws, derived from these models, accurately predict the transition boundaries observed in large-scale RNN training (new result, $\beta = 4.05 \pm 0.1$, $R^2 = 0.99$), when RNNs are learning those particular solutions in their latent subspaces.
>
> Here, it is worth noting that we empirically tested the scaling laws in the (realistic) full-rank RNNs, whereas the toy model theoretically models the latent trajectories in the low-rank RNN. The theoretical connection is an open question in this field and is a current hot topic (Valente et al., 2022). However, Valente et al. 2022 has empirically shown that trajectories of full-rank RNNs trained on behavioral tasks can often be reproduced by their low-rank counterparts that are sufficient to solve the task. In our case, rank-one RNN is sufficient to solve the task with a slow-point manifold, so it is not surprising that the toy model predictions explain (qualitatively) the existence of and (quantitatively) the boundary scalings of the phase diagram.
>
> *Longer rationale justifying, e.g., why the slow point model is a realistic approximation:*
> - RNNs are universal approximators of dynamical systems, and the rank of their recurrent weight matrix determines the dimensionality of the latent dynamics they can implement (Beiran et al., 2020).
> - Recent work has shown that even complex tasks can be solved with low-rank RNNs (Dubreuil et al., 2022), and that full-rank RNNs are often effectively low-rank in practice (Valente et al., 2022). In our empirical study, we similarly observe that both low- and full-rank RNNs converge to low-dimensional solutions, which take the form of either slow-point manifolds or limit cycles.
> - The slow-point toy model is the **normal form** (for reference, see Nonlinear Dynamics and Chaos by Strogatz) for a saddle-node bifurcation, which approximates one-dimensional dynamical systems around their local minimum (slow-point formation), including those implemented by the rank-one RNNs in Eq. (2).
> - Finally, learning the weights in rank-one RNNs implementing a slow-point corresponds to varying the parameter of this normal form, which gives rise to the phase diagrams derived in Fig 4C (and observed in full-rank RNNs in Fig 6).
>
> We will briefly incorporate this explanation into the revised manuscript, with the rigorous mathematical details in the supplementary.
>
> **(2)** We agree with the reviewer that *studying attractors as short-term memory mechanisms in RNNs is one of the oldest problems in computational neuroscience*. However, our contribution is not the problem itself, but our novelty lies in findings and mechanisms that were previously not known in this long-standing field. To be specific, here are a few of the novel findings in our work:
> - To our knowledge, this is the first study to systematically analyze how latent mechanisms emerge during learning (e.g., such a description does not exist for the STSP based models, also cited by the reviewer).
> - We identify a sharp transition between slow-point and limit-cycle solutions depending on task delay and learning rate. To our knowledge, we are the first to show that there is a phase diagram of strategies (w.r.t., optimization and task parameters).
> - Unlike prior work, which often focuses on complicated working memory tasks (e.g., saccades in the references cited by the referee), our custom task design focuses on short-term memory and balances simplicity and expressiveness, allowing us to analyze dynamical structures at scale.
> - We show that small changes (e.g, a post-response) can qualitatively shift the memory strategy learned.
> - Our large-scale dataset (35,000+ trained RNNs, first dataset of its size reported and to be made public) will be released to support further studies on training dynamics, robustness, and learning strategies.
>
> **Misc** We thank the reviewer for pointing out the omission of synaptic models of working memory, which we now explicitly cite. Please also see our response to Reviewer pHmu. We also see that you asked for an ethics review. How could we address your concerns?
>
> **Final remarks** We hope these clarifications address the reviewer’s concerns and are grateful for the feedback, which significantly improved the paper’s presentation and scope.

---

### Official Review · Reviewer_pHmu · 2025-03-14

**Overall Recommendation:** 4

**Summary:**

This paper studies computational RNN models of a classic neuroscience working memory task–the delayed response task–along with two, very simplified and tractable, dynamical system models capable of learning the task through adaptation of a scalar parameter. The paper studies the role that changes in the delay time, the response time, and an optional post-response period (where no output from the given model is expected) time play in how the models studied learn to solve the task. The main results of the paper is that RNNs tend to learn two different solutions to the task, one making use of a slow point the other a limit cycle, and that the solution learned will depend on the learning rate, the length of the delay and response periods, and the presence of a post-response period. It is observed that for longer delays and higher learning rates a limit cycle tends to be learned instead of a slow point solution. Interestingly, the loss function of a toy dynamical system model (the normal form of the saddle-node bifurcation) of learned slow-point dynamics requires a smaller learning rate or delay period to be learned, while a toy dynamical system for the limit cycle dynamics (sine function) is capable of learning with faster learning and longer delay periods. This relationship of better scaling w.r.t. learning rate and delay period for limit cycle solutions is proposed to underlie the phenomena observed in RNNS. Interestingly, the observed scaling in trained RNNs on the task without the post-response period is roughly what was predicted by the toy model. Lastly, for the numerical experiments, a very large sample of RNNs was trained and the authors plan to make these publicly available to facilitate future work.

The authors argue that their analysis of the learning of delay periods sheds light on the difficulty of learning long time dependencies in machine learning, and that it will help inform future neuroscience research by demonstrating how task parameters can have a significant effect on learning.

## Update After Rebuttal
I am satisfied by most of the authors clarifications and believe the paper is relevant to neuroscience and RNN-related learning and have thus updated my score to a 4. The remaining issue for me is that I still believe the authors could have done a better job connecting the toy models and the low-rank RNNs mathematically; this is why I have not rated the paper a strong accepted. Lastly, please disregard my mistaken replacement of $\dot{x}$ with $x$ in one of the comments.

**Claims And Evidence:**

In the reviewer’s view the claims seem well supported.

**Essential References Not Discussed:**

The reviewer is reasonably well versed with current neuroscience models of working memory (Working models of working memory – Barak & Tsodyks (2014) Current op. In Neurobiology).

The researcher is currently engaged in research on learning in the neuroscience-related firing-rate RNNs studied in the paper, and is therefore quite familiar with relevant literature, including:
- Flexible multitask computation in recurrent networks utilizes shared dynamical motifs – Driscoll et al. (2024) Nature Neuroscience
- Generating Coherent Patterns of Activity from Chaotic Neural Networks – Sussillo & Abbott (2009) Neuron
- A neuronal least-action principle for real-time learning in cortical circuits – Senn et al. (2024) eLife
- Partial observation can induce mechanistic mismatches in data-constrained models of neural dynamics – Qian (2024) biorXiv,

along with the papers mentioned above

**Experimental Designs Or Analyses:**

I did not check any of the code. The experimental designs seem appropriate for the questions that are being asked.

**Methods And Evaluation Criteria:**

Yes, they seem appropriate.

**Other Comments Or Suggestions:**

- In the abstract: the reviewer finds “limit cycles providing temporally localized approximations” a little confusing, and wonders if clarity could be increased for this sentence.
- In the abstract, the authors mention: “we derive theoretical scaling laws for critical learning rates as a function of the delay period length, beyond which no learning is possible.” It could be worth specifying that this derivation is in simplified dynamical system models rather than directly from RNNs
- Line 38, RHS: “studied” => studying
- Line 45, RHS: “periodicity” => periodic
- Line 73: LHS: I suggest: “Using interpretable dynamical system models stripped down to their most essential components for solving a delayed activation task” => “Using low-rank RNNs”. More concise and avoids confusion with analytical dynamical system models also studied in the paper
- Line 91, RHS: I suggest changing $W/W^{in}$ to $\{W, W^{in}\}$ to avoid notational confusion with division
- Equation 3: “$x$” => $\dot{x}$
- Line 359 RHS: exponents 2 and 3 should be -2 and -3.
- Line 663: “an halt” => a halt
- Eq S13: $T_{delay} >> T_{delay}$ => $T_{delay} >> T_{resp}$

**Other Strengths And Weaknesses:**

## Strengths
The paper does a fantastic job distilling a difficult problem down to a model that is simple enough for mathematical analysis but still seems to capture certain key aspects of the problem. The insights into two different strategies an RNN can use to solve a delay task, and how task structure and learning rate can influence these, are interesting and, in the reviewer’s view, worthy contributions.
## Weaknesses
The two main weaknesses, in the reviewer’s view, are: (1) that the simple dynamical systems studied are a substantial departure from an RNN and little justification is provided for the choice of simplified systems (Question 2, below); (2) how machine learning connections are primarily discussed in the discussion when the paper seems more relevant to computational neuroscience (Question 4, below); (3) the lack of discussion of certain alternative models of working memory (Question 1, below).

Note: the reason for the score is because the reviewer would like to see their comments and questions addressed before being able to recommend it for acceptance. If these are addressed satisfactorily the reviewer will certainly increase their score.

**Questions For Authors:**

1. To the reviewer’s understanding there is an alternative hypothesis for encoding working memory: that of short term plasticity (e.g. Working models of working memory – Barak & Tsodyks, 2014; see section on “Short term synaptic plasticity). Could the authors add mention of this hypothesis to the paper, or provide a compelling argument as to why it is irrelevant?
2. While compelling empirical evidence is provided for the relevance of the toy dynamical system models studied to learning in RNNs, the toy models are not rigorously connected to RNNs–at least beyond the sentence “It is worth noting that increasing the dimensionality of the dynamical system can allow more efficient solutions to the system, but the toy models we discuss can be thought as approximate bounds on what can be achieved.” Could the authors provide some deeper insight into this limitation, and the differences they expect between the behaviour of the toy models and of RNNs? This could be a useful limitation to include in the discussion.
3. The authors suggest that a key impact of their paper is the number of models the paper makes available. The reviewer wonders at the utility of providing 35,000 models all trained on a simple delay task. Could the authors propose different examples of ways this dataset could be used? The reviewer would also be curious about the environmental impacts of training this many models.
4. From the reviewer’s perspective this paper is primarily focused on learned computational mechanisms for solving a continuous-time delayed response task and therefore more relevant for neuroscience than machine learning. However, the authors spend the vast majority of the discussion talking about machine learning. The reviewer believes the paper would be more valuable if it spent more time discussing neuroscience applications, relevance to experimentalists, and potential hypotheses-to-test that the paper might generate; in particular, ways of distinguishing slow-point vs. limit cycle mechanisms experimentally. Would the authors be able to provide more discussion time on these neuroscience implications?
5. In biological networks one typically has modulatory signals that can increase or reduce network excitability. Do the authors have ideas for if and how a network with a limit cycle and excitability modulation (that can shut off the limit cycle after the response period) might be distinguished from the slow point mechanism, and how such a modulated limit cycle might compare with the two studied mechanisms?

**Relation To Broader Scientific Literature:**

There are several recent papers studying the “curse of memory” (difficulty learning long-timescales) in RNNs, the effect that long-memory tasks have on learning, and how to ameliorate it, that the reviewer believes could be relevant to cite:
- Approximation and Optimization Theory for Linear Continuous-Time Recurrent Neural Networks – Li et al. (2022) JMLR
- Recurrent neural networks: vanishing and exploding gradients are not the end of the story – Zucchet & Orvieto (2024) NeurIPS
- Generalized teacher forcing for learning chaotic dynamics – Hess et al. (2023) PMLR

The author also believes it could be useful to discuss models of working memory that rely on dynamics in variables other than firing rate, see for example the reference in Q1 of “Questions for Authors” section.

**Theoretical Claims:**

I checked the proofs in section S1.1 and they appear sound.

---

> ### Author Rebuttal · Authors · 2025-03-30
>
> We thank the reviewer for their thoughtful comments and highlighting our approach's strengths—especially our effort to simplify a complex problem into an interpretable framework. Their suggestions have greatly shaped our revisions. Below, we address all specific concerns and weaknesses.
>
> **Q1** That is an excellent point. As the reviewer notes, models based on dynamic synaptic variables like STSP represent a fundamentally distinct mechanism from the activity-based attractor models we focus on. In an earlier draft, we mentioned both classes—such as Mongillo et al. (2008)—and then narrowed the focus to activity-based models (Fig. 1). Based on the reviewer’s feedback, we restored and expanded the neuroscience context. In addition to intro, discussion mow highlights that STSP mechanisms may allow networks to bypass the learning constraints we identify for fixed-weight attractor models (e.g., Masse et al., 2019). The Conclusion links to recent AI models like Mamba, which integrate memory via local mechanisms that may functionally resemble STSP.
>
> **Q2** This point is addressed in our reply to Reviewer CYoj.
>
> **Q3** A key motivation for releasing all trained models is to enable reuse without retraining. We now clarify that the dataset required ~1 month of GPU time. Due to space constraints, we briefly outline three research directions this dataset enables (with more to be added in the Outlook):
>
> - Curriculum learning: Training pre-trained models for longer delays with reduced learning rates may enable faster convergence and offer insight into curriculum/transfer learning. The toy models could support new curriculum strategies with theoretical guarantees—relevant for experimentalists (including co-authors) who find it difficult to train mice on seconds-long delay STM tasks.
>
> - Model initialization: Exploring how weight initialization affects convergence to slow-point vs. limit-cycle strategies. For example, initializing near an existing solution (e.g., SP) may bias learning toward that strategy and potentially shift phase boundaries (Fig. 5).
>
> - Robustness: Comparing robustness by perturbing neurons (e.g., mimicking optogenetics) may reveal key differences (oscillations vs decay) between slow-point (SP) and limit cycle (LC) strategies.
>
> We also refer the reviewer to our response to Reviewer 59aj’s second question for more on the dataset’s utility.
>
> **Q4** Testable predictions. Our work predicts that both SP and LC dynamics can generate sequential neural activity, but differ in structure: SP dynamics converge to a slow-point attractor by trial end, whereas LC dynamics continue producing trial-type-specific sequences beyond reward. This distinction is testable by analyzing neural activity after task completion. While Rajan et al. (2016) explored sequences in RNNs, we link them to distinct attractor types. For instance, Fig. 4 predicts that extending the post-response period biases dynamics toward LC (oscillatory) or SP (ramping) solutions. Although this period can’t be fully removed in experiments, it can be disrupted—e.g., via bulk optogenetic inhibition of task-relevant regions, providing a plausible way to test this prediction.
>
> Relation to recent studies: We will cite the suggested works and place greater emphasis on neuroscience connections (previously omitted due to space). The reviewer’s interest—as a neuroscientist—strongly encouraged this expansion.-
>
> **Q5**  We thank the reviewer for raising this thought-provoking possibility of a third dynamic regime—one in which a network operates in a limit-cycle (LC) mode during the task but undergoes a modulatory shift in excitability toward the end of the trial that disrupts the remaining cycle, effectively mimicking a slow-point (SP) solution in the end. This is indeed how we would expect LC solutions to be utilized in practice. Distinguishing such a modulated LC from a true SP is an exciting experimental challenge and could reveal important aspects of synaptic learning rules and neuromodulatory control. We outline three levels of experimental inquiry to test this hypothesis in vivo (briefly):
> - Exp 1: The assumption of functional reorganization can be tested by combining constant-power bulk optogenetic activation with - calcium imaging at different trial stages.
> - Exp 2: Genetically encoded fluorescent sensors (e.g., dLight, GRAB-ACh) can be used to track neuromodulatory signals during task performance. A signal peaking in the post-response window would support the proposed mechanism.
> - Exp 3: To establish causal roles, one could block or genetically delete the relevant receptors and observe changes in post-trial neural dynamics or task performance.
>
> **Final remarks** Thank you for the style suggestions and pointing out the typos. We incorporated all suggested edits and reviewed the manuscript for clarity, consistency, and polish. Please let us know if any points need clarification. We sincerely appreciate the reviewer’s enthusiasm and attention to detail.

---

> > ### Comment · Reviewer_pHmu · 2025-04-03
> >
> > Thank you very much for the comprehensive response!
> >
> > I have a few more clarifying questions (numbering does not correspond exactly to the previous question numbers):
> >
> > 1. regarding your response to reviewer CYoj on the connection between toy model and theory: I agree that modelling RNNs with low-rank RNNs is entirely justified. My question about connecting toy model and theory is not about connecting RNN and low-rank RNN, but rather drawing a connection between the low-rank RNN and the toy models used. For example, would it be possible to demonstrate how the saddle node normal form could be derived from a rank 1 RNN, and how the parameters of the RNN might relate to the parameter $r$? For the $\dot{x} = \sin(2\pi r t)$ model, perhaps you could provide similar intuition for how it would relate to the 2D low rank RNN, and how its $r$ parameter might relate?
> >
> > 2. Thank you very much for the extra details on GPU use for the dataset! As mentioned in my original review, given the month-long compute time for the project, it would be great to have a bit more information on the environmental impact. For example, it could be beneficial to include (at least in the supplementary) an estimate of both GPU hours used and GPU type, along with an approximate estimate of carbon emissions. This should not be too difficult, as emissions calculators are available online--see e.g.: https://mlco2.github.io/impact/
> >
> > 3. In your response to **Q4** above you suggest that LC and SP could be distinguished by bulk optogenetic inhibition. How would the effect of opto inhibition on SP differ from its effect on LC?
> >
> > 4. To clarify, in your response to **Q4** above you mention "*Fig. 4 predicts that extending the post-response period biases dynamics toward LC (oscillatory) or SP (ramping) solutions*". Do you not mean that it biases towards LC and away from SP solutions?
> >
> > 5. In your response to **Q4** above you mention "*This distinction is testable by analyzing neural activity after task completion.*" Wouldn't a mechanism like the one I imagined in **Q5** from my review make it difficult to distinguish LC and SP based on activity after task completion?
> >
> > 6. Thank you very much for the detailed response to **Q5**. Will you mention this as a potential confound for distinguishing LC and SP in the main text?

---

> > > ### Author Response · Authors · 2025-04-03
> > >
> > > Thank you for the additional questions. Please find our answers below:
> > >
> > > 1) Please allow us to elaborate with step by step mathematical derivations:
> > >
> > > - A low-rank RNN *universally* approximates a flow map, $\dot \kappa (t) = G(\kappa(t))$, when we train its parameters during learning.
> > >
> > > - Any flow map (consider 1-d case), unless the second derivative vanishes, can be approximated by a Taylor series around its minimum (e.g. where slow point's locus is), where the first derivative vanishes due to the extremum conditions. Hence, the flow map becomes $\dot \kappa(t) = a + b \kappa(t)^2 + O(\kappa^3)$.
> > >
> > > - Here, we can simply rescale the time (using the symmetry there) to fix $b=1$ without loss of generality, achieving the normal form. This is a standard calculation in dynamical system literature and would amount to rescaling n and m in our RNN such that their multiplication W = mn^T remains invariant (e.g., an inherent symmetry of the system).
> > >
> > > - Then, for a given set of n and m variables that support this slow-point manifold, the latent dynamical system approximated by the rank-one RNN would approximate this function such that $\dot \kappa(t) = - \kappa(t) +n^T \tanh(m\kappa(t)) :\approx r + \kappa^2$.
> > >
> > > - Now, since there is an attractor around some $\kappa \approx \kappa^*$, the latent activities will be in the neighborhood of the local minima (again, by definition of an attractor) most of the time, hence the time evolution of the RNN will be well approximated by this local approximation (also shown in our Fig. S1).
> > >
> > > -  During learning, the small changes in parameter n (a similar argument can be made for changes in m) would then be approximated as: $(n+Δn)^T \tanh((m)\kappa) = n^T tanh(m\kappa) + Δn^T \tanh(m\kappa)$. Since $\kappa(t) \approx \kappa^*$ most of the time (which becomes exact as $T \to \infty$ as slow point becomes a fixed point), the second term can be approximated ny $ Δn^T \tanh(m\kappa^*)$. This is a linear function of $Δ n$, i.e., change in the encoding weights.
> > >
> > > - Hence, the gradient that governs $r$ in the toy model $\dot \kappa = \kappa^2 + r$ also governs the infinitesimally linear changes,  $Δ n$ corresponds to $r$ in the toy model! But then, this is already enough to make our case, since changes in $Δ n$ will be subject to the same scaling laws as $r$ in our toy model did in the limit $T \to \infty$.
> > >
> > > Now, for the limit cycle, we can follow exactly the same arguments, but we have to assume that latent dynamical system is two dimensional. Moreover, we would need to use the form in Eq. (S15) not (S16) as the former is the dynamical system not the latter, though the steps for the derivations are analogous (where we now approximately fix the radius $||\kappa||_2 = R*$). In the end, the math comes down to $\dot \kappa_1 = - 2 \pi r \kappa_2$, where changes in $r$ would correspond to $Δm$.
> > >
> > > 2) Thank you for this great website. We used computers with NVIDA RTX 3090 GPUs, which amounts to about 110 kg CO2 emission, 440 km driven by an ICE car, 55 kg coal burned, about 2 tree seedlings sequesting carbon for 10 year. We will acknowledge this and cite the website in the acknowledgement. This is an important point, thank you!
> > >
> > > 3) Bulk optogenetic during trials would disrupt the neural activities in both models outside of the steady-state. For a limit cycle, the return to steady-state would be in the form continuing the sequential activity, whereas slow-point manifolds that define sequential activity through closeness to the locus would "restart" the sequence. This would be a first evidence, though experiments in Q5 would need to be conducted. Bulk optogenetic post-trial could allow us to test whether we can bias the solutions as predicted by theory, since when bulk opto is applied during learning, animals could not reliably use post-trial window for learning.
> > > 4) Correct, this was a typo.
> > > 5) Exactly, this is why the reviewer's intuition was correct and the experiment in Q5 is a necessary next step. On the other hand, if animals were forced to refrain their responses after their behavior for a short window before their reward is delivered, it could also be expected to see echoes of a limit cycle (which is how we plan to start these experiments with mice). Notably, if limit cycles are observed after the trial completion, that is a definite evidence; whereas during trial, limit cycles, slow-points, and an intermediary mechanism described by the reviewer, would behave similarly during trial.
> > >
> > > 6) Yes, we will include these in the discussion, which is in the main text. Specifically, discussion will now be more tailored for neuroscientists.
> > >
> > > We believe this was our final response due to the rules. As an experimental neuroscience lab, we are confident we can present these points clearly and accessibly in the final version. We are grateful for the reviewer’s engagement and hope our responses support an improved evaluation!

---

### Decision · Program_Chairs · 2025-05-01

**Decision:**

Accept (poster)

**Comment:**

The paper investigates the strategies being used by RNNs to maintain short-term memories and identified two mechanisms of slow-point manifolds and limit cycles. The paper has done thorough analysis of the trained network and how it is dependent on training parameters. Reviewers appreciate the distillation of a complex problem down to a simple model that is plausible for math analysis. Although there are concerns from reviewers (missing citations and discussions, weak connection to experimental neuroscience, etc), overall, I think contribution of the study outweighs the weakness and decide to accept this paper. Please revise the manuscript based on reviewers’ suggestions.